



**Field assessments on impact of $CO_2$ concentration fluctuations along with complex**
**terrain flows on the estimation of the net ecosystem exchange of temperate forests**
Dexiong Teng[1,2], Jiaojun Zhu[1,2,3,*], Tian Gao[1,2,3], Fengyuan Yu[1,2], Yuan Zhu[1,2], Xinhua
Zhou[3,4], Bai Yang[4]
1 Institute of Applied Ecology, Chinese Academy of Sciences, Shenyang 110000, China
2 Qingyuan Forest CERN, National Observation and Research Station, Liaoning Province,
Shenyang 110016, China
3 CAS-CSI Joint Laboratory of Research and Development for Monitoring Forest Fluxes of Trace
Gases and Isotope Elements, Institute of Applied Ecology, Chinese Academy of Sciences,
Shenyang 110016, China
4 Campbell Scientific Incorporation, Logan, Utah 84321, USA
**\* Corresponding Author: Jiaojun Zhu**
Tel : +86 24 83970342
Email: jiaojunzhu@iae.ac.cn





## Abstract

The $CO_2$ storage ($F_s$) is the cumulation or depletion in $CO_2$ amount over a period in an ecosystem. Along with the eddy-covariance flux and wind-stream advection of $CO_2$, it is a major term in the net ecosystem $CO_2$ exchange (NEE) equation and even dominates in the equation under a stable atmospheric stratification while this equation is used for forest ecosystems over complex terrains. However, estimating the $F_s$ remains challenging due to the frequent gusts and random fluctuations in boundary-layer flows that arouse tremendous difficulties in catching the true trend of $CO_2$ changes for its storage estimation from eddy-covariance along with the atmospheric profile techniques. Using the measurements from Qingyuan Ker Towers equipped with NEE instrument systems separately covering mixed-broadleaf, oak, and larch forests towers in a mountain watershed, this study investigates the gust periods and $CO_2$ fluctuation magnitudes while examining their impact on $F_s$ estimation in relation to the terrain complexity index (TCI). The gusts induce $CO_2$ fluctuations at numerous periods of 1 to 10 min over two hours. Diurnal, seasonal, and spatial differences ($P < 0.01$) in the maximum amplitude of $CO_2$ fluctuations ($A_{max}$) ranges from 1.6 to 136.7 ppm and these difference in a period ($P_{max}$) at the same significant level ranges 140 to 170 second. The $A_{max}$ and $P_{max}$ are significantly correlated to the magnitude and random error of $F_s$ with diurnal and seasonal differences. These correlations decrease as $CO_2$ averaging time windows becomes longer. To minimize the uncertainties of $F_s$, a constant $[CO_2]$ averaging time window for the $F_s$ estimates is not ideal. Dynamic averaging time windows and a decision-level fusion model can reduce the potential underestimation of





$F_s$ by 29%–33%, being equivalent to 1.9%–4.3% underestimation of the NEE for
temperate forests in complex terrains. The relative contribution of $F_s$ to the 30-min NEE
observations ranged from 17% to 82% depending on wind speed and TCI. The study's
approach is notable as it incorporates TCI and utilizes three flux towers for replication,
making the findings relevant to similar regions with a single tower.
**Keywords**: Eddy covariance, complex terrain, carbon flux, storage term, carbon
dioxide concentration, random uncertainty
**1 Introduction**

The accurate estimation of the net ecosystem exchange (NEE) of carbon dioxide

($CO_2$) in forest ecosystems is crucial for a comprehensive understanding of the global
carbon cycle. The eddy covariance (EC) technique has been widely used in forest
ecosystems due to its capacity to directly measure the NEE while measurement
conditions satisfy the underlying theory. The EC technique is based on a simplified
mass conservation equation (after the Reynolds averaging), given by:

$$
\begin{aligned}
\text{NEE} = &\underbrace{\frac{1}{V_m}\int_0^h\left(\frac{\partial \bar{c}}{\partial t}\right)dz}_{\text{I}} + \underbrace{\frac{1}{V_m}\left(\overline{w'c'}\right)_h}_{\text{II}} \\
&+ \frac{1}{V_m}\int_0^h\left(\underbrace{\bar{w}(z)\frac{\partial \bar{c}}{\partial z}}_{\text{III}a} + \underbrace{\bar{c}(z)\frac{\partial \bar{w}}{\partial z}}_{\text{III}b}\right)dz, \\
&+ \frac{1}{V_m}\int_0^h\underbrace{\left(\bar{u}(z)\frac{\partial \bar{c}}{\partial x} + \bar{v}(z)\frac{\partial \bar{c}}{\partial y}\right)}_{\text{IV}}dz
\end{aligned}
\tag{1}
$$


where $V_m$ is the volume of dry air in the control volume; $c$ is the $CO_2$ mixing ratio; $t$ is
the time; $h$ is the measure height; $u$, $v$, and $w$ denote the velocity components in the $x$,





$y$, and $z$ directions, respectively; and an overbar denotes Reynolds averaging. This
equation conceptualizes the NEE within a control volume from the ground to the
measurement height ($h$), while ignoring the horizontal turbulence term divergence
(Feigenwinter et al., 2004). In this equation, term I is the $CO_2$ storage ($F_s$) representing
the change in the average $CO_2$ concentration (hereafter $[CO_2]$). Terms II, IIIa, IIIb, and
IV represent the vertical turbulent flux ($F_c$), the vertical advection, the interface vertical
mass advection, such as the evaporation process (Webb et al., 1980), and the horizontal
advection, respectively.

Most flux measurements typically lack the solutions for terms III and IV, and can

only estimate the NEE by summing $F_c$ and $F_s$, and even a significant number of sites
ignored the $F_s$. The $F_s$ in the vertical gas column within a canopy can be substantial,
requiring attention in NEE estimates (Aubinet et al., 2000). The $F_s$ contributes ~60% to
nocturnal turbulent flux underestimation in forest ecosystems with "ideal" topography
(Mchugh et al., 2017). Especially, during atmospherically stable periods such as the
early morning, sunset, and nighttime transitions, the $F_s$ has a significant impact on the
NEE. For 30-min and annual forest ecosystem carbon flux measurements, ignoring $F_s$
would underestimate the NEE (Zhang et al., 2010). The $F_s$ value typically ranges from
$-2$ to $-5$ μmol m$^{-2}$ s$^{-1}$ in the early morning, and the $F_s$ is about 1–3 μmol m$^{-2}$ s$^{-1}$ after
sunset for temperate forests. Neglecting the $F_s$ value can also lead to a misunderstanding
of the $CO_2$ exchange processes, such as ecosystem respiration and photosynthesis, and
their relationship with key control factors such as solar radiation, temperature, and
moisture (Mchugh et al., 2017). Therefore, it is imperative not to overlook $F_s$ to ensure



more precise NEE estimates of forest ecosystems, particularly in complex terrains.

Despite the challenges inherent in monitoring forest conditions, understanding the

carbon flux of forest ecosystems in complex terrains or with heterogeneous underlying
surfaces remains an area of great interest. Topography complexity plays a complex role
in the transportation of momentum, energy, and mass in the atmospheric boundary layer,
with direct impacts on the airflow patterns, spatiotemporal characteristics, and gas
concentration fluctuations (Sha et al., 2021; Finnigan et al., 2020). Differences in
airflow along the slope, lateral $CO_2$ discharge downhill, and spatiotemporal variations
in soil respiration result in the $CO_2$ outflow from slopes and valleys lagging behind the
flat top of the mountain (De Araújo et al., 2010). At night, under stable atmospheric
stratification, cold air moves from the ground to the valley forest canopy and then flows
to low-lying areas, causing a "carbon pooling" effect. The gradient of $[CO_2]$ below the
EC sensors fluctuates significantly, and the cold air discharge above the canopy reduces
$CO_2$ storage, leading to an underestimation of forest ecosystem respiration (Yao et al.,
2011; De Araújo et al., 2008; De Araújo et al., 2010).

According to the theoretical definition, $F_s$ estimates are derived by averaging the

$[CO_2]$ of the control volume at the beginning and the end of the EC averaging period
(30 min or 1 h) and dividing by the EC averaging period (Finnigan, 2006). In practice,
the $F_s$ represents the integration of the time derivative of the vertically determined
column-averaged $[CO_2]$. However, relying solely on tower-top measurements can lead
to underestimation of $F_s$ by up to 34% compared to the eight-level profile approach (Gu
et al., 2012). The NEE magnitude with the $F_s$ based on the two-min $[CO_2]$ averaging



time window (instantaneous concentration approach) was found to be 5% higher than
that of the 30-min-window-based $F_s$ (averaging concentration approach), particularly
during nighttime in the growing season (Wang et al., 2016). The effect of the $F_s$ on the
NEE of forest ecosystems increases with the increase of timescale, and the annual sum
of the NEE obtained using the instantaneous concentration approach is higher than that
obtained by averaging concentrations (Li et al., 2020). Most research has examined how
vertical and horizontal gas concentration sampling point distribution affects the
uncertainty in $F_s$ estimation (Bjorkegren et al., 2015; Wang et al., 2016; Yang et al.,
2007; Yang et al., 1999), with a small number of studies examining the effect of $[CO_2]$
sampling frequency on the $F_s$ (Finnigan, 2006; Heinesch et al., 2007). Certain studies
have experimentally validated new concepts, such as correlating the gas sampling point
concentration with the horizontal distribution (Nicolini et al., 2018). Some studies have
approached the true value theoretically, such as through defining the control volume
represented by flux measurements (Metzger, 2018; Xu et al., 2019). However, the
number of complete column samples required to describe the column-averaged $[CO_2]$
of each 30-min or 1-h $F_s$ estimate is still undetermined.

Previous studies have emphasized the significance of the $F_s$ to the NEE and the

influence of $[CO_2]$ dynamics on $F_s$ estimates in complex terrains. To overcome any
disparities between sensors and obtain precise changes in the $[CO_2]$ gradient above and
below the forest canopy, individual gas analyzers are extensively utilized to measure
$[CO_2]$ levels vertically (Siebicke et al., 2011). However, a single gas analyzer introduces
time delays when monitoring multi-point $[CO_2]$ curves. Accurately determining the $F_s$



estimates can be challenging due to the spatial and temporal resolution of $[CO_2]$
measurements (Wang et al., 2016). The random error of the $F_s$ estimates using one
complete column sample is considerably high due to short-term $[CO_2]$ fluctuations
(Nicolini et al., 2018). The calculation of the $F_s$ using time-averaged $[CO_2]$ profiling
leads to significant information loss at high frequency, resulting in a substantial
underestimation bias. Furthermore, resource constraints in the measurement system,
coupled with a lack of clear guidelines for estimating the $F_s$ values and their associated
uncertainties, create a significant gap between ideal initiatives and their implementation.
These issues necessitate further efforts to characterize $[CO_2]$ fluctuations across
different sites and demonstrate the mechanisms influencing $F_s$ magnitudes,
uncertainties, and their contributions to NEE observations in complex terrains. Thus,
this manuscript aims to bridge this gap by introducing a statistical method to estimate
$F_s$ values and their uncertainties.

This paper employed an innovative EC site with three flux towers (Qingyuan-Ker

Towers) to monitor three typical types of temperate forest stands located in complex
terrains in northeastern China. This study introduces a decision-level fusion model
based on weighing the underestimation bias and random error of the $F_s$ to obtain more
accurate results. The objectives of this study were to: 1) compare diurnal, seasonal, and
spatial differences in $[CO_2]$ fluctuations, $F_s$, and its uncertainty; 2) examine the
variation in $F_s$ uncertainty with different $[CO_2]$ averaging time windows; and 3)
investigate the response of $F_s$ and its uncertainty to $[CO_2]$ fluctuations, wind above the
canopy, and terrain complexity, and quantify the impact of the $F_s$ on the NEE estimates



under these conditions.

## 2 Materials and methods

2.1 Study site and instrumental set-up


This study was conducted in temperate forests in a watershed based on the Ker
towers (Zhu et al., 2021; Gao et al., 2020), situated in northeast China (41°50′N,
124°56′E). The region experiences a temperate continental monsoon climate, with an
average annual temperature of 4.3 °C and annual rainfall of 758 mm from 2010 to 2021
(Li et al., 2023). The Ker towers consist of three 50-m-high EC towers (Fig. 1) that
observe a mixed broadleaved forest (MBF), a Mongolian oak forest (MOF), and a Larch
plantation forest (LPF).

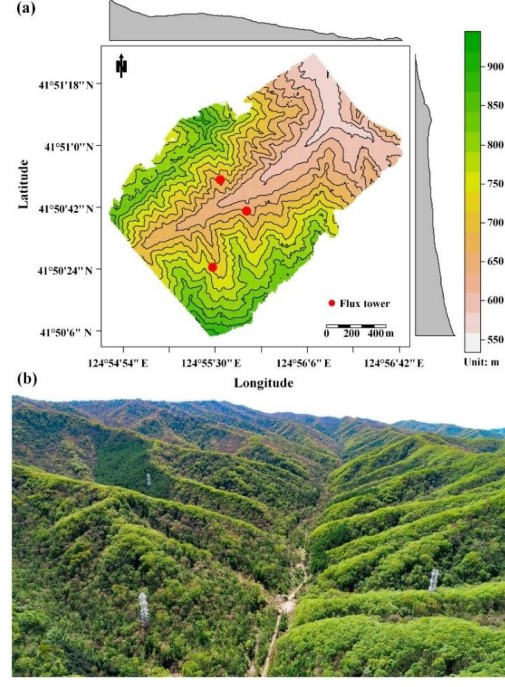


Fig. 1 Overview of the study area. The first map (a) depicts the topography of the study site, with



black curves indicating elevation contours, and marginal distributions represented as a gray graph,

averaged over rows and columns. The second image (b) features an aerial photograph of the

Qingyuan-Ker towers captured in the growing season (Gao et al., 2020).

The basic information regarding Ker towers in this study is presented in Table 1.

The closed-path EC system (EC310, Campbell Scientific Ltd., Logan, UT, USA),
comprising a CSAT3 sonic anemometer and an EC155 closed-path infrared ray gas
analyzer (IRGA), was employed to monitor the three-dimensional wind speed and
$CO_2/H_2O$ concentrations (10 Hz). The atmospheric profiling system (AP200, Campbell
Scientific Ltd., Logan, UT, USA) was utilized to measure the $CO_2/H_2O$ concentrations
with eight height levels. Each level was measured for 15 s (with 10 s for the flushing
of the manifold and 5 s for logging the average), leading to a measurement cycle of 2
min.
Table 1 Basic information of Ker towers

| Forest | Mixed broad-leaved | Mongolian oak | Larch plantation |
|---|---|---|---|
| Experiment period | Jan 01, 2020–Dec 31, 2021 | Jan 01, 2020–Dec 31, 2021 | Jan 01, 2020–Dec 31, 2021 |
| Elevation (m) | 634 | 669 | 721 |
| Slope (°) | $14.8 \pm 2.1$ | $19.1 \pm 2.9$ | $16.2 \pm 5.3$ |
| Canopy height (m) | $21.5 \pm 1.8$ | $13.9 \pm 0.6$ | $19.5 \pm 0.6$ |
| Leaf area indices | $3.0 \pm 0.5$ | $3.1 \pm 0.8$ | $3.9 \pm 0.6$ |
| Eddy covariance system | EC310 | EC310 | EC310 |
| Eddy covariance sensor height (m) | 46 | 46 | 36 |
| Atmospheric profiling system | AP200 | AP200 | AP200 |
| Profile heights (m) | 0.5, 2, 6, 11, 16, 21, 26, 36 | 0.5, 2, 6, 11, 16, 21, 26, 36 | 0.5, 2, 6, 11, 16, 21, 26, 36 |



Atmospheric
Measurement
Techniques



Discussions

2.2 Calculation of storage flux
Averaging the [CO₂] in a time window was utilized to calculate the $F_s$ values, in
addition to data on the air pressure, $CO_2/H_2O$ molar fractions, and air temperature at
different heights above the ground surface (Finnigan, 2006; Montagnani et al., 2018;
Xu et al., 2019). The molar mixing ratio and mass mixing ratio are conserved quantities
with the variation of air temperature, air pressure, and water vapor concentration,
whereas the molar fraction is not. This study determined the $F_s$ using the molar mixing
ratio obtained from $CO_2/H_2O$ molar fraction observations, applying the ideal gas law
and Dolton's partial pressure law (Montagnani et al., 2009). The water vapor molar
mixing ratio $(\chi_v)$ in mmol mol$^{-1}$ is given by

$$\chi_v = \frac{c_v}{1 - c_v \times 10^{-3}}, \tag{2}$$

where $c_v$ is the water vapor molar fraction in mmol mol$^{-1}$, and the $CO_2$ molar mixing
ratio $(\chi_c)$ in μmol mol$^{-1}$ is given by

$$\chi_c = \frac{c_c}{1 - c_v \times 10^{-3}}, \tag{3}$$

where $c_c$ is the $CO_2$ molar fraction in μmol mol$^{-1}$.
The dry air density $(\bar{\rho}_d)$ in mol m$^{-3}$ is calculated as follows:

$$\bar{\rho}_d = \frac{\bar{P}}{(\bar{T} + 273.15) \times (R^* + \chi_v \times 10^{-3} \cdot R^* \cdot M_d/M_v)}, \tag{4}$$

where $R^*$ is the air gas constant (8.31441 Pa m$^3$ K$^{-1}$ mol$^{-1}$), $\bar{P}$ is the air pressure in
Pa, and $\bar{T}$ is the average air temperature in Celsius. $M_d$ and $M_v$ are the dry air and
water vapor molar mass (18.015 g mol$^{-1}$), respectively. $M_d$ is calculated from the $CO_2$
molar mixing ratio (Khélifa et al., 2007):





$$M_d = 28.9635 + M_c \cdot (\chi_c \times 10^{-6} - 0.0004), \qquad (5)$$

where $M_c$ is the CO₂ molar mass (12.011 g mol⁻¹).
The $F_s$ estimated from eight-level profiles are calculated as follows:

$$F_s = \bar{\rho}_d \int_0^h \frac{d\bar{\chi}_c}{dt} dz \doteq \bar{\rho}_d \sum_{i=1}^{8} \frac{\Delta\bar{\chi}_{c_i}\Delta h_i}{\Delta t}, \qquad (6)$$

where $\bar{\chi}_c$ is the average CO₂ molar mixing ratio and $\Delta h_i$ is the height represented by
each level. To ensure that $F_s$ corresponds to $F_c$ in time, the average [CO₂] at the start or
end moments (*t*) during a time window (*τ* min) is calculated as follows:

$$\bar{\chi}_{c_i} = \frac{2}{\tau} \sum_{t-\frac{\tau}{2}<t\leq t+\frac{\tau}{2}} \chi_{c_i}(t). \qquad (7)$$

2.3 Data analysis
To evaluate the impact of [CO₂] fluctuations on $F_s$ measurements and its
corresponding uncertainty, empirical modal decomposition (EMD) and Fourier
spectrum analysis were used to extract the period and amplitude of fluctuations in the
high-frequency [CO₂] time series (10 Hz). EMD was used to decompose the [CO₂] time
series into intrinsic mode functions based on local signal properties, which yield
instantaneous frequencies as functions of time, allowing for the identification of
embedded structures of eddies. EMD is applicable to non-linear and non-stationary
processes (Huang et al., 1998). The period and amplitude of [CO₂] fluctuations above
the forest canopies reflected the eddy size. Subsequently, the maximum period and
amplitude of [CO₂] fluctuations in a short term (2h) was indicative of large eddies under
the influence of gust.
Due to the diurnal and seasonal variability of flux measurements, this study





defined the transition period and growing season. The solar elevation angle was used
to define the transition period as 1-h before sunrise (sunset) to 2-h after sunrise (sunset).
The growing degree days (GDDs) were calculated using the base temperature ($T_{base}$) to
determine the beginning and end of the growing season, and the formula was as follows
(Mcmaster and Wilhelm, 1997):

$$GDD = \frac{1}{2}(T_{max} + T_{min}) - T_{base}, \qquad (8)$$

where $T_{base}$ is 6°C. Considering the persistent demand of temperature to support
vegetation growth, the fourth day of the first GDD greater than zero (less than zero)
over a span of five consecutive days was defined as the starting (ending) time of the
growing season.

The main data processing and analysis steps are outlined below:

1. EMD and Fourier spectrum analysis of [$CO_2$] high-frequency time series were

used to extract the maximum amplitude ($A_{max}$) and corresponding period ($P_{max}$) of [$CO_2$]
fluctuations every 2 h. The data were divided into two subsets based on $P_{max}$, with a
cut-off of 150 s.

2. $CO_2$ storage fluxes were calculated for different [$CO_2$] average time windows

($\tau$), ranging from 4 to 28 min.

3. The standardized major axis (SMA) regression model (Warton et al., 2012) was

used to compare the slope differences (bias) between $F_{s\_\tau}$ and $F_{s\_28}$ for different $P_{max}$
and the forest stands. The SMA model offers routines for comparing parameters $a$ and
$b$ among groups for symmetric problems.

4. The normalized root mean square error (NRMSE) and slope were used to



evaluate the relative error and bias between $F_{s\_\tau}$ and $F_{s\_28}$. The NRMSE is calculated as
follows:

$$NRMSE = 100 \times \sqrt{\frac{\sum_{i=1}^{N}(F_{s\_\tau}^{(i)} - F_{s\_28}^{(i)})^2}{\sum_{i=1}^{N}(F_{s\_28}^{(i)} - \overline{F_{s\_28}})^2}} \qquad (9)$$

5. The normalized weighting coefficient ($w$) of $F_{s\_\tau}$ was estimated based on the
NRMSE and slope (Wang et al., 2020). The details are shown in Appendix A1. Then,
using the decision-level fusion model, $F_{s\_comb}$ was calculated as follows:

$$F_{s\_comb} = w_1^* \cdot F_{s\_4} + w_2^* \cdot F_{s\_8} + \cdots + w_7^* \cdot F_{s\_28} \qquad (10)$$

The decision-level fusion model automatically assigned weights to the $F_s$ based on
different $[CO_2]$ averaging time windows. Its purpose in this study was to balance the
relative error and bias of $F_s$ estimates caused by $[CO_2]$ sampling. The analysis was
performed using the EMD and smatr R packages (Warton et al., 2012; Huang et al.,

1998).

2.4 Uncertainty analysis
To improve the accuracy of estimating the uncertainty of $F_s$ using individual tower,
this work has made modifications to the 24-h difference method by extending the
sampling time windows and applying meteorological condition constraints (Hollinger
and Richardson, 2005). This method trades time for space to estimate the uncertainty
of $F_s$. To determine the uncertainty of $F_s$, this study compared the observations at
moment $i$ to the average of several observations during a similar period and with similar
meteorological conditions. The specific steps were as follows:
1. The average $F_s$ was calculated in a certain time window (15 d) for the moment
interval ($i-0.5$ h, $i+0.5$ h) where moment $i$ was located and where the meteorological
conditions (such as the $u_*$, air temperature, and sensible heat flux) were similar.
2. The difference between the $F_s$ value corresponding to each moment $i$ and $\overline{F_s}$
was calculated separately to obtain the residual sequence $\varepsilon_s$.
3. The standard deviation $\sigma(\varepsilon_s)$ related to $\varepsilon_s$ for $F_s$ was calculated in a certain
time window (15 d) for the moment interval ($i-0.5$ h, $i+0.5$ h) where moment $i$ is located
and where the meteorological conditions (such as the $u_*$, air temperature, and sensible
heat flux) were similar.
After estimating the uncertainty of $F_s$, this study extended the work conducted by
Richardson et al. (2008) to analyze its relationship with the magnitude of flux
measurements ($|F_s|$), [$CO_2$] fluctuations ($A_{max}$ and $P_{max}$), wind speed (WS), and terrain
complexity index (TCI). A comprehensible description of the TCI can be found in
Appendix A2. This relationship can be approximated by using the following equation:

$$\sigma(\varepsilon_s) = \beta_0 + \sum_{i=1} \beta_i \cdot x_i, \tag{11}$$

where the nonzero intercept term $\beta_0$ indicates the size of the random uncertainty as
$x_i$ approaches 0, which varies with the observation site, with larger value of $\beta_0$
indicating greater uncertainty. The slope term $\beta_i$ indicates the sensitivity of the size of
the random uncertainty of $x_i$, with smaller $\beta_i$ values indicating a probability
distribution of uncertainty closer to white noise.

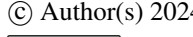



**3 Results**

3.1 Characterization of [$CO_2$] fluctuation and $F_s$ variations

The [$CO_2$] high-frequency time series above the forest canopies were decomposed using EMD, followed by spectral analysis to extract the fluctuation period and amplitude of [$CO_2$] at different time scales. As depicted in Fig. 2, it became evident that the [$CO_2$] above the canopies displayed short-term fluctuations with periods ranging from 1 to 10 min, and the amplitude of these fluctuations showed an increasing trend with longer periods. This observation strongly suggested the presence of large eddies influenced by gusts above the canopies, and these eddies were responsible for the increasing amplitude of [$CO_2$] fluctuations as their size increased.

To examine the spatio-temporal variations in large eddies, this study compared the $A_{max}$ and $P_{max}$ values above canopies across different forest stands. The analysis utilized data from daytime, nighttime, and transition periods in both the growing and dormant seasons. The averages of $A_{max}$ and $P_{max}$ averages for the above-canopy [$CO_2$] in the three forest stands ranged from 1.588 to 136.667 ppm and from 2.313 to 2.784 min, respectively (Table 2). Fig. 3 demonstrated significant seasonal and diurnal differences ($P < 0.01$) in $P_{max}$, with higher values during daytime in the growing season, and lower values during the daytime in the dormant season. Moreover, $P_{max}$ was significantly different ($P < 0.01$) among different forest stands during the same time period, with MBF stand having the highest values, followed by the MOF, and the lowest values in the LPF. During the growing season, the $A_{max}$ values were significantly higher than





those during the dormant season, with both daytime and nighttime values also
exhibiting significant differences ($P < 0.01$) among different forest stands. This
observation provided evidence of significant spatio-temporal variability in large eddies
influenced by gusts.

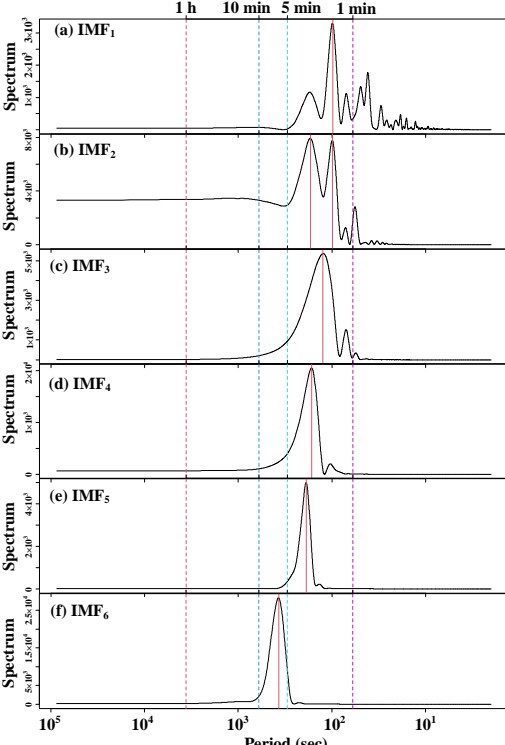


Fig. 2 Power spectral density of the intrinsic mode function (IMF) of above-canopy $CO_2$
concentrations in the Mongolian oak forest on July 2, 2020 (24 h).
Table 2 Mean of the $A_{max}$ and $P_{max}$ in different forest stands at different periods

| Variable | Tower site | Growing season | | | Dormant season | | |
|---|---|---|---|---|---|---|---|
| | | DT[1] | NT[2] | TP[3] | DT | NT | TP |
| $A_{max}$[4] (ppm) | MBF[6] | 57.932 | **139.667** | 136.717 | 2.219 | 5.212 | 4.944 |
| | MOF[7] | 36.160 | 57.945 | 55.777 | 2.699 | 5.175 | 4.637 |
| | LPF[8] | 52.688 | 58.816 | 60.147 | **1.588** | 2.985 | 2.456 |
| $P_{max}$[5] | MBF | 154.563 | **167.024** | 164.824 | 158.449 | 151.428 | 158.121 |



| (s) | MOF | 151.986 | 160.633 | 159.146 | | 153.091 | 147.491 | 153.274 |
| | LPF | 149.003 | 143.950 | 145.696 | | 143.458 | **138.794** | 142.009 |

[1] DT represents daytime; [2] NT represents nighttime; [3] TP represents transition period. [4] $A_{max}$ represents the maximum amplitude of short-term $CO_2$ concentration fluctuations; [5] $P_{max}$ represents the corresponding period of maximum amplitude. [6] MBF represents mixed broad-leaved forest; [7] MOF represents Mongolian oak forest; [8] LPF represents Larch plantation forest.

To estimate the uncertainty of $F_s$ using an individual tower, a comprehensive analysis of its diurnal and seasonal dynamics, as well as the functional relationship between $F_s$ and $u_*$, was necessary. Fig. 4 presented significant diurnal variations and seasonal differences in $F_s$ across the three forest stands. During the growing season, the median diurnal variation of $F_s$ for the three forest stands ranged from $-2.960$ to $2.647$ $\mu$mol m$^{-2}$ s$^{-1}$, whereas during the dormant season, it ranged from $-1.306$ to $1.012$ $\mu$mol m$^{-2}$ s$^{-1}$. Comparing the amplitude of $F_s$ diurnal variation among the three forest stands, MBF exhibited the largest amplitude during the growing season, while the amplitudes of the three forest stands were similar during the dormant season. Notably, it was observed that the amplitudes for longer $P_{max}$ values were greater than those for shorter $P_{max}$ values. This observation indicated that the larger the eddies, the greater the amplitude of $F_s$.

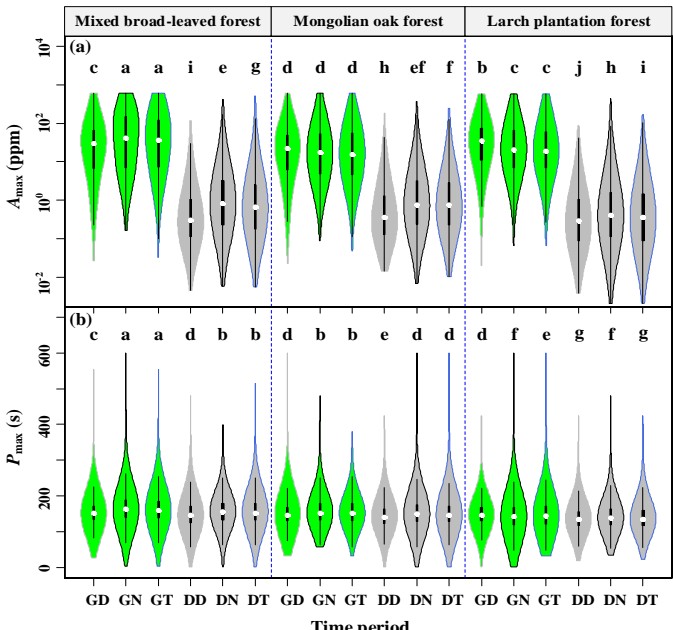

Fig. 3 Maximum amplitude ($A_{max}$) (a) and corresponding period ($P_{max}$) (b) of short-term $CO_2$

concentration fluctuations in different forest stands for seasonal and diurnal variations, where GD,

GN, GT, DD, DN, and DT denote the growing season daytime, growing season nighttime,

growing season transition period, dormant season daytime, dormant season nighttime, and

dormant season transition period, respectively. Columns with different lowercase letters are

significantly different ($P < 0.05$) according to Fisher's least significant difference test.

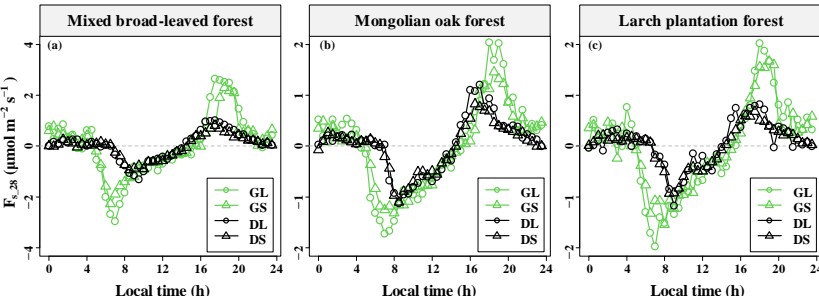

Fig. 4 Median diurnal variation of $CO_2$ storage flux ($F_s$) based on 28-min $CO_2$ concentration

averaging time windows in the three forest stands during different seasons. GS indicates the

growing season and a short period of maximum amplitude ($P_{max}$), GL indicates the growing

season and a long $P_{max}$, DS indicates the dormant season and a short $P_{max}$, and DL indicates the

dormant season and a long $P_{max}$.

Furthermore, a $u_*$ threshold value was identified for the variation of $F_s$ with $u_*$


during daytime in both the dormant and growing seasons (Fig. 5). When $u_*$ fell below
the $u_*$ threshold, the magnitude of $F_s$ ($|F_s|$) decreased with increasing $u_*$. Conversely,
when $u_*$ exceeded the $u_*$ threshold, the $|F_s|$ tended to remain relatively constant. Notably,
a maximum point for the $|F_s|$ was observed when the $u_*$ was less than 0.5 m/s during the
growing season, whereas not during the dormant season. This phenomenon was
particularly evident during the nighttime and transition periods of the growing season,
where $|F_s|$ exhibited an initial increase followed by a subsequent decrease with $u_*$. These
observations strongly indicated that the effect of the turbulent mixing strength on the
$|F_s|$ over complex terrains was nonlinear and exhibited diurnal and seasonal differences.

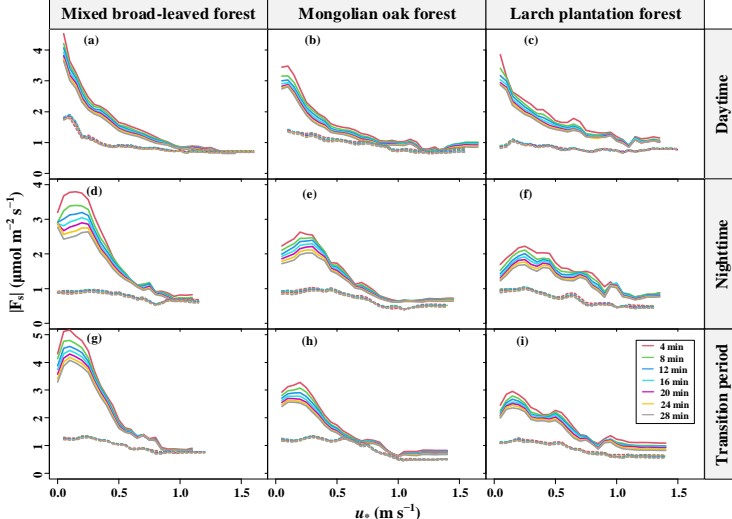


Fig. 5 Magnitudes of $CO_2$ storage flux ($|F_s|$) determined with different $CO_2$ concentration average
time windows as a function of the friction velocity ($u_*$) and moving block averages from all 30-
min data for the years 2020-2021. Dashed and solid lines donate the dormant and growing
seasons, respectively.
3.2 Effect of $[CO_2]$ fluctuations on the $F_s$ and its uncertainty

To investigate the influence of the $[CO_2]$ fluctuation periods on the error of $F_s$





measurement, this study computed the diurnal average of the standard deviation $\sigma(\varepsilon_s)$
of the 30-min $F_s$ uncertainty ($\varepsilon_s$) separately for different $P_{max}$ values and the seasons.
The overall distribution of $\varepsilon_s$ showed a non-normal distribution with a high peak
(kurtosis > 2 and $P < 0.05$, results presented in Supplementary Table 1–4). The daily
variation curves of $\sigma(\varepsilon_s)$ at various [$CO_2$] averaging time windows are presented in
Fig. 6. It was observed that the diurnal variation range of $\sigma(\varepsilon_s)$ was higher during the
growing season compared to the dormant season, regardless of the $P_{max}$ lengths,
indicating a seasonal difference independent of the $P_{max}$. Additionally, during the
growing season, both MBF and MOF demonstrated evident diurnal variation in $\sigma(\varepsilon_s)$,
with the peak occurring at night and the trough during the daytime. The diurnal
variation range of $\sigma(\varepsilon_s)$ varied across the three forest stands, with MBF exhibiting the
largest amplitude.

Furthermore, a significantly positive correlation was observed between $\sigma(\varepsilon_s)$ the

|$F_s$| ($P < 0.01$), with site, seasonal, and diurnal differences (Fig. 7). The relationship
between these variables was characterized by intercepts and slopes that varied across
different [$CO_2$] averaging time windows, ranging from 1.99 to 2.82 and from 0.24 to
0.28, respectively (results presented in the Supplementary Tables 5–6). Both decreased
as the [$CO_2$] averaging time window increased, with the growing season exhibiting
larger values compared to the dormant season (results shown in the Supplementary
Tables 5–6). These findings suggested that increasing the [$CO_2$] averaging time window,
results in a reduction of the random error in $F_s$, approaching a behavior similar to white
noise.



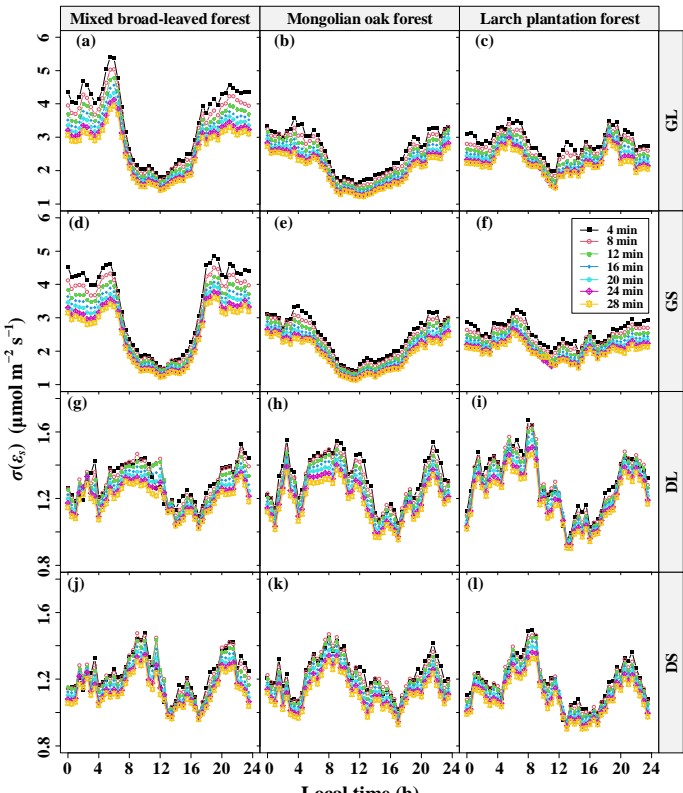

Fig. 6 Diurnal variations in the random uncertainty ($\sigma(\varepsilon_s)$) of $CO_2$ storage flux ($F_s$) errors ($\varepsilon_s$) at different $CO_2$ concentration ([$CO_2$]) averaging time windows and their seasonal differences, where GS indicates the growing season and a short period of maximum amplitude ($P_{max}$) of [$CO_2$] fluctuations, GL indicates the growing season and a long $P_{max}$, DS indicates the dormant season and a short $P_{max}$, and DL indicates the dormant season and a long $P_{max}$.

To assess the impact of [$CO_2$] fluctuations on the error and bias of $F_s$ measurement, this study compared the NRMSE and slopes of $F_s$ based on different [$CO_2$] averaging time windows, with reference to the baseline $F_{s\_28}$, across various $P_{max}$ values, time periods, and sites. As shown in Fig. 8, the NRMSE decreased and approached convergence as the [$CO_2$] averaging time windows increased. During both daytime and nighttime in the growing season, the NRMSE corresponding to longer $P_{max}$ was greater than that corresponding to shorter $P_{max}$, while the opposite trend was observed during



the dormant season. Additionally, the longer the $[CO_2]$ averaging time window, the
greater the relative underestimation of $F_s$.

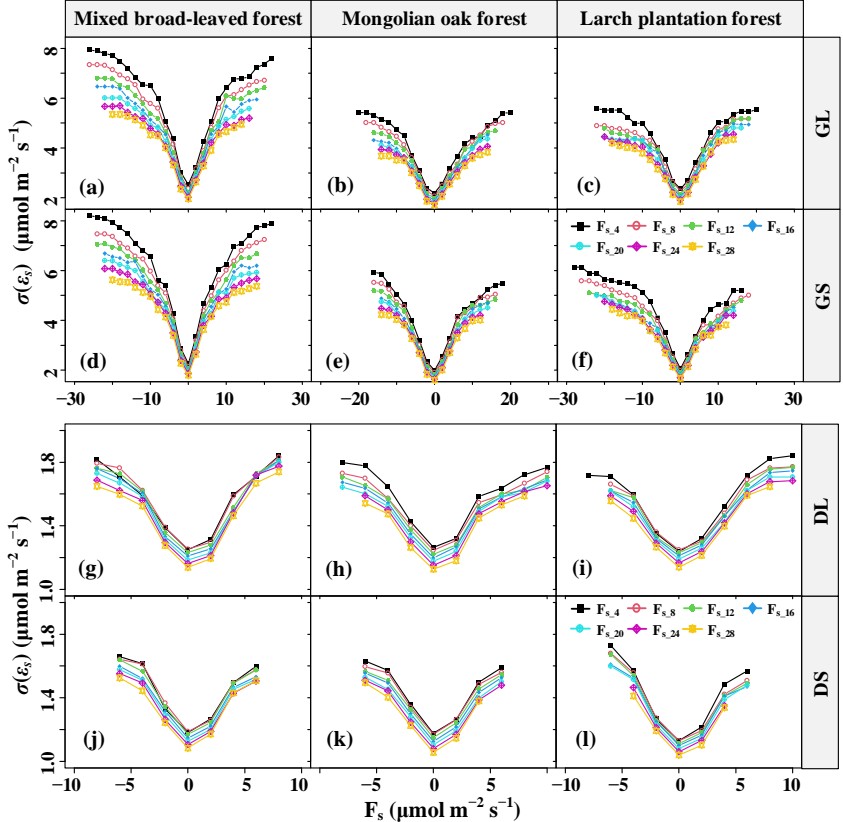


Fig. 7 Random uncertainty $\sigma(\varepsilon_s)$ of $CO_2$ storage flux ($F_s$) errors ($\varepsilon_s$) at different $CO_2$
concentration ($[CO_2]$) averaging time windows as a function of the $F_s$ magnitude for mixed broad-
leaved forest, Mongolian oak forest, and Larch plantation forest during the growing and dormant
seasons. GS indicates the growing season and a short period of maximum amplitude ($P_{max}$) of
$[CO_2]$ fluctuations, GL indicates the growing season and a long $P_{max}$, DS indicates the dormant
season and a short $P_{max}$, and DL indicates the dormant season and a long $P_{max}$.





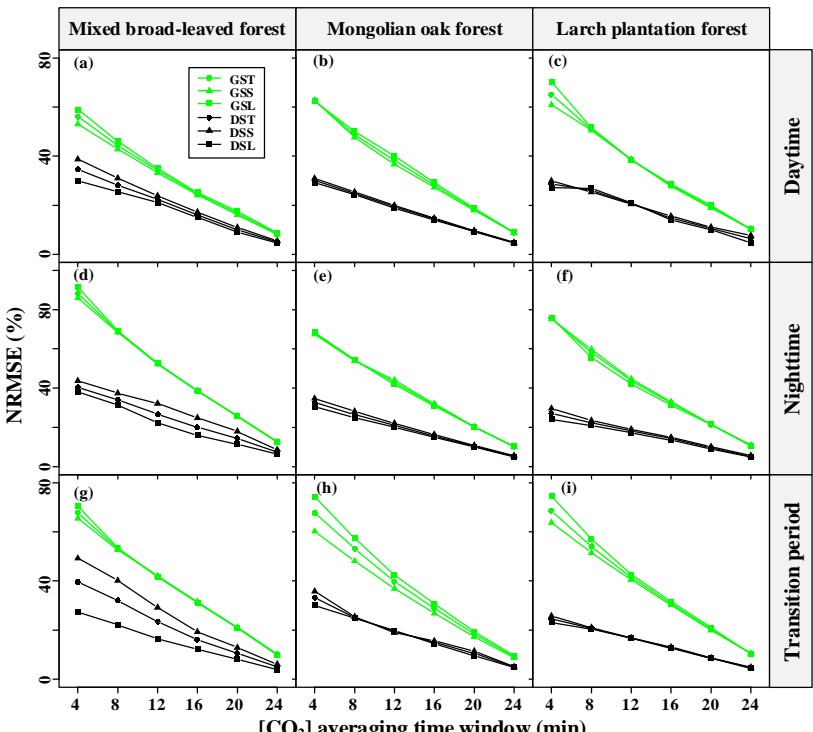


Fig. 8 Seasonal and diurnal differences in the normalized root mean square error (NRMSE) of

$CO_2$ storage flux ($F_s$) versus the respective $F_{s\_28}$ values for different $CO_2$ concentration ([$CO_2$])

averaging time windows. GST indicates the growing season and does not distinguish the period of

maximum amplitude ($P_{max}$) of [$CO_2$] fluctuations, GSS indicates the growing season and a short

$P_{max}$, GSL indicates the growing season and a long $P_{max}$, DST indicates the dormant season and

does not distinguish $P_{max}$, DSS indicates the dormant season and a short $P_{max}$, and DSL indicates

the dormant season and a long $P_{max}$.

The comparison of slopes between $F_{s\_4}$ and $F_{s\_28}$ in the three forest stands revealed

interesting patterns, as depicted in Fig. 9. During the growing season, the slopes

corresponding to the shorter $P_{max}$ of [$CO_2$] fluctuations were consistently lower than

those for the longer $P_{max}$, indicating that the effect of $P_{max}$ on $F_s$ uncertainty decreased

with increasing [$CO_2$] averaging time windows. However, for the MBF stand (Fig. 9d

and Fig. 9g), the slopes corresponding to the shorter $P_{max}$ of [$CO_2$] fluctuations during

the dormant season nighttime were actually greater than those for the longer $P_{max}$,





primarily due to diurnal variations in the daily dynamics of $F_s$. Overall, the influence of
$P_{max}$ on $F_s$ uncertainty decreased with increasing $[CO_2]$ averaging time windows. This
suggested that averaging $[CO_2]$ reduced the effect of gusts on the random uncertainty
in estimating $F_s$, but led to a systematic underestimation of $F_s$.

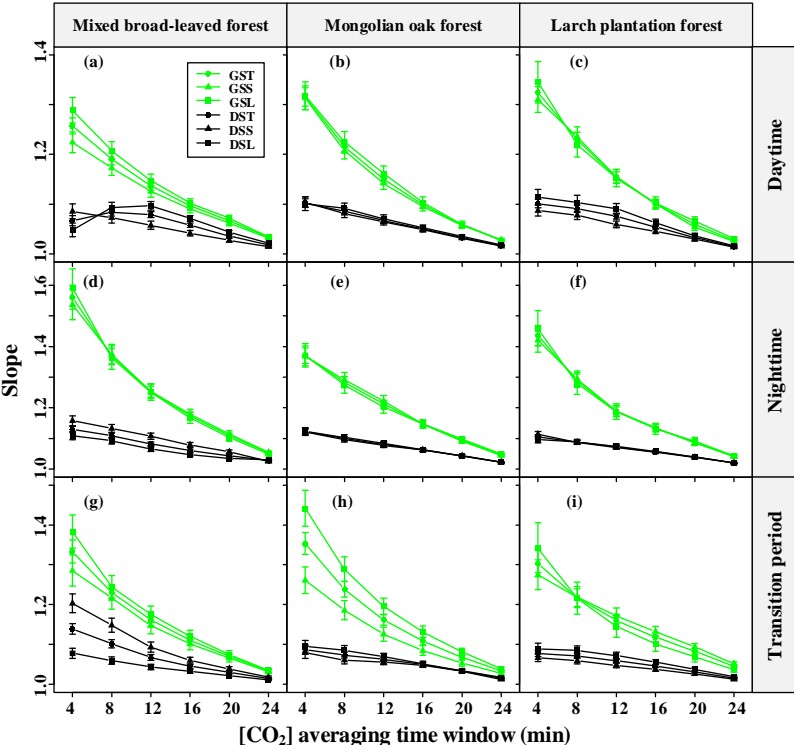


Fig. 9 Seasonal and diurnal differences in the slope of $CO_2$ storage flux ($F_s$) versus the $F_{s\_28}$ for the
different $CO_2$ concentration ($[CO_2]$) averaging time windows. GST indicates the growing season
and does not distinguish the period of maximum amplitude ($P_{max}$) cases, GSS indicates the
growing season and a short $P_{max}$, GSL indicates the growing season and a long $P_{max}$, DST
indicates the dormant season and does not distinguish $P_{max}$, DSS indicates the dormant season and
a short $P_{max}$, and DSL indicates the dormant season and a long $P_{max}$.

To analyze the effect of $[CO_2]$ fluctuations on $|F_s|$ in complex terrains, this study

developed a multiple linear regression model, considering the interaction effects of
wind speed and terrain complexity on $|F_s|$, as shown in Fig. 10. $A_{max}$ exhibited a





significant positive correlation with $|F_s|$ in all time periods ($P < 0.05$). Conversely, $P_{max}$
showed a significant negative correlation with $|F_s|$ during the dormant season daytime,
the growing season daytime, and the transition periods ($P < 0.05$). Additionally, their
correlation coefficient decreased with increasing $\tau$. In Fig. 10d and Fig. 10e, a $u_*$
threshold was observed during the growing season nighttime. When the $u_*$ was below
the threshold, higher TCI values resulted in smaller $|F_s|$; whereas when the $u_*$ was above
the threshold, higher TCI values led to larger $|F_s|$. During the growing season nighttime
and transition periods, $u_*$ showed a significant negative correlation ($P < 0.05$) with $|F_s|$,
and the correlation coefficient decreased with increasing TCI values.

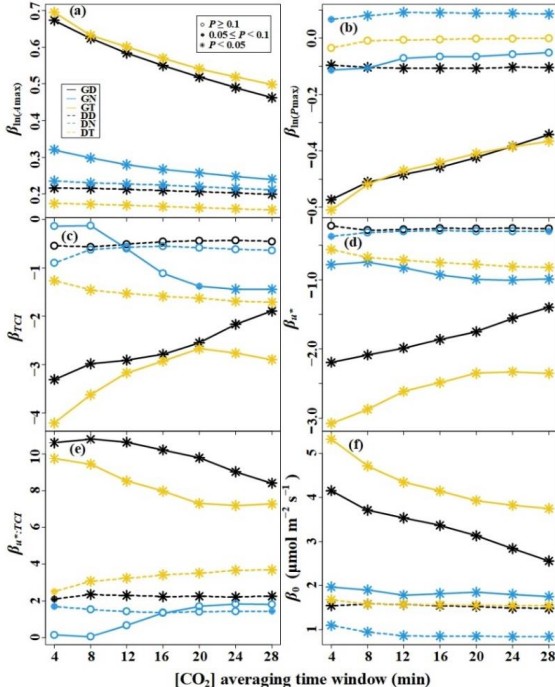


Fig. 10 Linear regression coefficients of the $CO_2$ storage flux ($F_s$) magnitude—driving factors
relationships for the seven $CO_2$ concentration ($[CO_2]$) averaging time windows. $u_*$: friction
velocity; TCI: terrain complexity index; $A_{max}$: maximum amplitude of $[CO_2]$ fluctuations; $P_{max}$:
corresponding period of maximum amplitude.



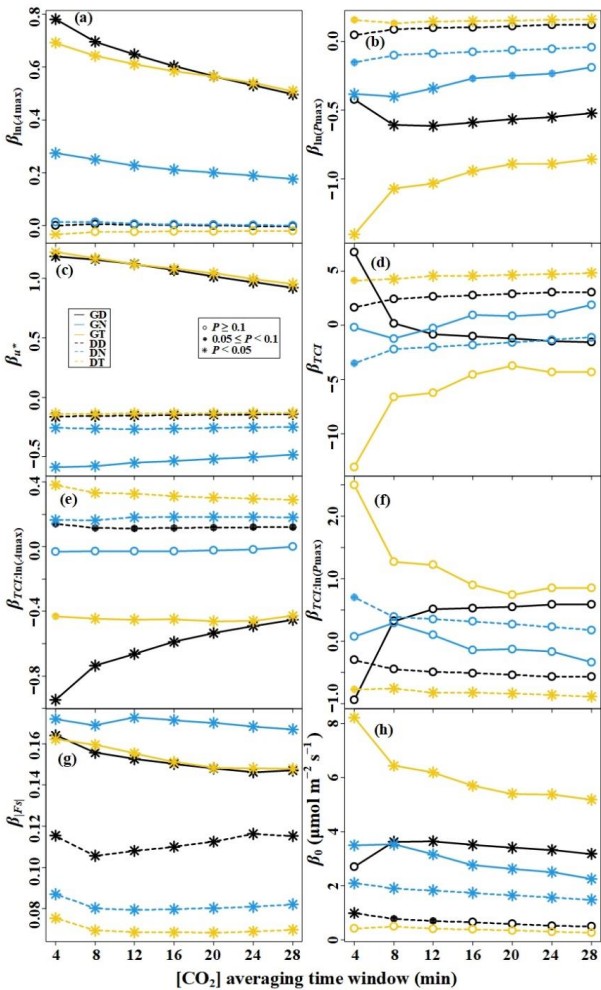


Fig. 11 Linear regression coefficients of the standard deviation of $CO_2$ storage flux ($F_s$)—driving

factors relationships determined with Eq. (11) for the seven $CO_2$ concentration ([$CO_2$]) averaging

time windows. $u*$: friction velocity; TCI: terrain complexity index; $A_{max}$: maximum amplitude of

[$CO_2$] fluctuations; $P_{max}$: corresponding period of maximum amplitude.

As evident from Fig. 11a and Fig. 11e, the $A_{max}$ exhibited a significant positive

correlation ($P < 0.05$) with $\sigma(\varepsilon_s)$ during both the dormant season's nighttime and the

growing season. Throughout the transition period of the growing season, $P_{max}$ displayed

a significant negative correlation with $\sigma(\varepsilon_s)$ ($P < 0.05$). During the transition period

of the dormant season, a TCI threshold was observed, with $P_{max}$ showing a significant





positive correlation ($P < 0.05$) with $\sigma(\varepsilon_s)$ when the TCI was below the threshold, and
a significantly negative correlation ($P < 0.05$) with $\sigma(\varepsilon_s)$ when the TCI exceeded the
threshold (Fig. 11b and Fig. 11f). The $u_*$ showed a significantly negative correlation
with $\sigma(\varepsilon_s)$ during the daytime and transition periods of the growing season ($P < 0.05$),
while in other time periods, $u_*$ was significantly positively correlated with $\sigma(\varepsilon_s)$ ($P <$
$0.05$). The $|F_s|$ demonstrated a significant positive correlation with $\sigma(\varepsilon_s)$ ($P < 0.05$) in
all time periods, with its correlation coefficient being greater during the growing season
than during the dormant season. These observations suggested that the effect of
turbulent mixing on the magnitude of $F_s$ and its uncertainty was regulated by terrain
complexity.
3.3 Effect of $CO_2$ storage fluxes uncertainty on NEE observations

The 30-min $F_{s\_comb}$ was obtained by weighing the bias and random error of $F_s$ using

different $[CO_2]$ averaging time windows and $P_{max}$ values. This study then focused on
the magnitude of $F_{s\_comb}$ in relation to the $F_c$ magnitude and its diurnal, seasonal, and
site variations. To assess the significance of $F_s$ in NEE observations, the relative
contribution ratio of $F_{s\_comb}$ magnitude ($|F_{s\_comb}|/(|F_c|+|F_{s\_comb}|)$) was employed. The
$|F_{s\_comb}|/(|F_c|+|F_{s\_comb}|)$ showed a decreasing trend to convergence with increasing $u_*$
(Fig. 12). On average, the $|F_{s\_comb}|/(|F_c|+|F_{s\_comb}|)$ ranged from 17.2% to 82.0%, with a
higher value during the dormant season compared to the growing season. This indicated
that as turbulence intensity increased, the contribution of $F_s$ to the NEE in forests
decreased to a constant value. Nevertheless, even under strong turbulence intensity, $F_s$
still played a significant role in the NEE observations of forests in complex terrains.



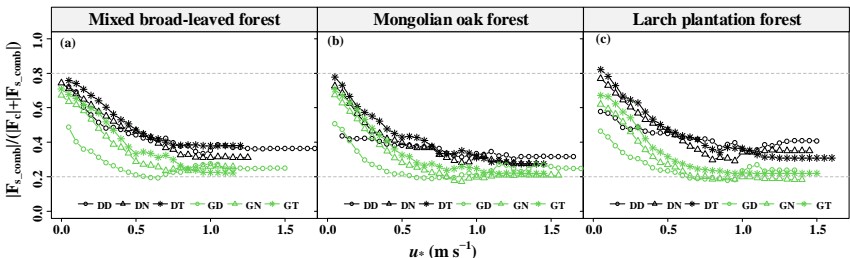

Fig. 12 Relative contribution ratio of the $CO_2$ storage flux magnitude ($|F_{s\_comb}|/(|F_c|+|F_{s\_comb}|)$) determined by decision-level fusion model as a function of the friction velocity ($u_*$) moving block averages from all 30-min data for the years 2020–2021. GD represents the growing season's daytime; GN represents the growing season's nighttime; GT represents the growing season's transition period; DD represents the dormant season's daytime; DN represents the dormant season's nighttime; DT represents the dormant season's transition period.

As indicated in Table 3, both $P_{max}$ and TCI exhibited a significant positive correlation with $|F_{s\_comb}|/(|F_c|+|F_{s\_comb}|)$ ($P < 0.05$), while both $A_{max}$ and WS showed a significant negative correlation with $|F_{s\_comb}|/(|F_c|+|F_{s\_comb}|)$ ($P < 0.05$). Notably, seasonal variations in correlation coefficients were observed. The correlation between the WS and $|F_{s\_comb}|/(|F_c|+|F_{s\_comb}|)$ was more pronounced during both the dormant season's transition period and the growing season, and it decreased with increasing TCI values during the dormant season's daytime and nighttime.

Table 4 presented a comprehensive comparison of diurnal, seasonal, and site differences in slope (with intercept terms forced to zero) and NRMSE between $F_{s\_comb}$ and $F_{s\_28}$. The slopes for the three forest stands ranged from 28.6% to 33.3%, with higher slopes observed during the growing season compared to the dormant season. This suggested that the $F_{s\_28}$ was underestimated by 28.6%–33.3% compared to the $F_{s\_comb}$. The NRMSE of $F_{s\_comb}$ versus the $F_{s\_28}$ in the three forest stands ranged from 59.2% to 67.2%.





Table 3 Linear regression coefficients of the relative contribution ratio of $F_{s\_comb}$
magnitudes to NEE observations ($|F_{s\_comb}|/(|F_c|+|F_{s\_comb}|)$) —driving factors
relationships for the six time periods.

| Time period | $\beta_0$ | $\ln(P_{\max})$[7] | $\ln(A_{\max})$[8] | $u_*$[9] | TCI[10] | $u_*$:TCI | $R^2$ |
|---|---|---|---|---|---|---|---|
| Total | 0.292 *** | 0.048 *** | −0.037 *** | −0.334 *** | 0.790 *** | −1.018 *** | 0.278 *** |
| GD[1] | 0.299 *** | 0.016 | −0.041 *** | −0.183 *** | −0.293 * | 0.239 | 0.158 *** |
| GN[2] | 0.370 *** | 0.029 | −0.023 *** | −0.386 *** | −0.038 | 0.081 | 0.103 *** |
| GT[3] | 0.161 | 0.060 *** | −0.014 *** | −0.182 | 1.056 *** | −1.754 | 0.186 *** |
| DD[4] | 0.393 *** | 0.011 | −0.020 *** | −0.154 * | 0.306 | −0.153 | 0.040 *** |
| DN[5] | 0.661 *** | 0.012 | −0.026 *** | −0.443 *** | −0.035 | 0.399 | 0.088 *** |
| DT[6] | 0.495 *** | 0.017 | −0.036 *** | −0.294 *** | 0.564 | −0.852 | 0.149 *** |

[1] GD represents the growing season's daytime; [2] GN represents the growing season's nighttime;
[3] GT represents the growing season's transition period; [4] DD represents the dormant season's
daytime; [5] DN represents the dormant season's nighttime; [6] DT represents the dormant season's
transition period. [7] $A_{max}$: maximum amplitude; [8] $P_{max}$: corresponding period of maximum amplitude.
[9] $u_*$: friction velocity; [10] TCI: terrain complexity index; *** represents $P < 0.001$; ** represents $P <$
0.01; * represents $P < 0.05$.
To evaluate the impact of $F_{s\_comb}$ on $NEE_{obs}$ ($F_c + F_s$), we further evaluated the
slope (with intercept terms forced to zero) and NRMSE of $F_c + F_{s\_comb}$ compared to $F_c$
$+ F_{s\_28}$, as presented in Table 5. The slopes for the three forest stands indicated that the
$NEE_{obs}$ with $F_{s\_28}$ was underestimated by 1.9%–4.3% compared to the $NEE_{obs}$ with
$F_{s\_comb}$, with the order MBF > LPF > MOF. The underestimation of $NEE_{obs}$ was higher
(6.4%–15.5%) during the growing season nighttime and transition period but much





lower (< 1%) during the growing season daytime. The NRMSE of $NEE_{obs}$ with the
$F_{s\_comb}$ versus the $F_{s\_28}$ in the three forest stands ranged from 16.0% to 25.4%. The
analysis suggested that combining the $F_s$ values based on different averaging $[CO_2]$
time windows in the decision-level fusion model could successfully weigh potential
underestimation bias and random uncertainties.
Table 4 Statistical inference of major axis regression for the $F_{s\_28}$ calculated by
combining multiple $[CO_2]$ averaging time windows $F_{s\_comb}$ and the 28-min averaging
window-based $F_s$

| Forest | Time period | N | $R^2$ | Slope | 95% CI | | NRMSE % |
|---|---|---|---|---|---|---|---|
| MBF[7] | Total | 28061 | 0.729 | **1.333** | 1.323 | 1.342 | **67.2** |
| | GD[1] | 5726 | 0.813 | 1.248 | 1.234 | 1.262 | 55.1 |
| | GN[2] | 3640 | 0.643 | 1.535 | 1.503 | 1.567 | 85.8 |
| | GT[3] | 3092 | 0.725 | 1.311 | 1.287 | 1.337 | 67.2 |
| | DD[4] | 4906 | 0.914 | 1.064 | 1.054 | 1.075 | 32.9 |
| | DN[5] | 6791 | 0.900 | 1.099 | 1.089 | 1.109 | 35.0 |
| | DT[6] | 3906 | 0.948 | 1.068 | 1.059 | 1.077 | 25.0 |
| MOF[8] | Total | 28817 | 0.783 | **1.286** | 1.279 | 1.294 | **59.2** |
| | GD | 5886 | 0.781 | 1.281 | 1.266 | 1.296 | 61.0 |
| | GN | 3799 | 0.752 | 1.338 | 1.318 | 1.358 | 65.6 |
| | GT | 3190 | 0.751 | 1.350 | 1.327 | 1.373 | 66.8 |
| | DD | 5236 | 0.931 | 1.108 | 1.098 | 1.118 | 31.1 |
| | DN | 6669 | 0.934 | 1.110 | 1.103 | 1.118 | 29.3 |
| | DT | 4037 | 0.951 | 1.079 | 1.070 | 1.088 | 24.7 |
| LPF[9] | Total | 24273 | 0.763 | **1.287** | 1.278 | 1.296 | **61.4** |
| | GD | 4659 | 0.732 | 1.335 | 1.314 | 1.357 | 68.2 |
| | GN | 2995 | 0.733 | 1.444 | 1.418 | 1.472 | 73.7 |
| | GT | 2544 | 0.737 | 1.250 | 1.225 | 1.28 | 63.1 |
| | DD | 4231 | 0.937 | 1.104 | 1.094 | 1.113 | 29.3 |
| | DN | 6288 | 0.952 | 1.094 | 1.088 | 1.100 | 24.8 |
| | DT | 3556 | 0.963 | 1.071 | 1.064 | 1.079 | 21.2 |





[1] GD represents the growing season's daytime; [2] GN represents the growing season's nighttime;
[3] GT represents the growing season's transition period; [4] DD represents the dormant season's
daytime; [5] DN represents the dormant season's nighttime; [6] DT represents the dormant season's
transition period. [7] MBF represents mixed broad-leaved forest; [8] MOF represents Mongolian oak
forest; [9] LPF represents Larch plantation forest.
Table 5 Statistical inference of major axis regression for the $F_c+F_{s\_28}$ (NEE$_{obs}$)
calculated by combining multiple $[CO_2]$ averaging time windows ($F_c+F_{s\_comb}$) and the
28-min averaging window-based NEE$_{obs}$

| Forest | Time period | N | $R^2$ | Slope | 95% CI | | NRMSE % |
|---|---|---|---|---|---|---|---|
| MBF[7] | Total | 28061 | 0.942 | **1.043** | 1.040 | 1.046 | 25.4 |
| | GD[1] | 5726 | 0.986 | 1.010 | 1.007 | 1.014 | 15.9 |
| | GN[2] | 3640 | 0.828 | 1.155 | 1.138 | 1.172 | 50.8 |
| | GT[3] | 3092 | 0.839 | 1.136 | 1.119 | 1.154 | 46.3 |
| | DD[4] | 4906 | 0.984 | 1.010 | 1.007 | 1.014 | 12.8 |
| | DN[5] | 6791 | 0.949 | 1.040 | 1.034 | 1.045 | 23.9 |
| | DT[6] | 3906 | 0.949 | 1.050 | 1.043 | 1.058 | 23.9 |
| MOF[8] | Total | 28817 | 0.976 | **1.019** | 1.017 | 1.020 | 16.0 |
| | GD | 5886 | 0.993 | 1.005 | 1.002 | 1.007 | 11.7 |
| | GN | 3799 | 0.908 | 1.078 | 1.067 | 1.089 | 39.1 |
| | GT | 3190 | 0.892 | 1.097 | 1.084 | 1.110 | 38.3 |
| | DD | 5236 | 0.993 | 1.010 | 1.008 | 1.012 | 8.6 |
| | DN | 6669 | 0.974 | 1.033 | 1.029 | 1.037 | 17.1 |
| | DT | 4037 | 0.972 | 1.022 | 1.016 | 1.027 | 17.7 |
| LPF[9] | Total | 24273 | 0.969 | **1.024** | 1.021 | 1.026 | 18.1 |
| | GD | 4659 | 0.984 | 1.012 | 1.008 | 1.016 | 17.0 |
| | GN | 2995 | 0.938 | 1.062 | 1.052 | 1.072 | 31.9 |
| | GT | 2544 | 0.891 | 1.064 | 1.050 | 1.079 | 38.1 |
| | DD | 4231 | 0.989 | 1.017 | 1.014 | 1.020 | 10.9 |
| | DN | 6288 | 0.979 | 1.035 | 1.031 | 1.038 | 15.5 |
| | DT | 3556 | 0.980 | 1.030 | 1.025 | 1.035 | 14.8 |

[1] GD represents the growing season daytime; [2] GN represents the growing season nighttime; [3]
GT represents the growing season transition period; [4] DD represents the dormant season daytime; [5]
DN represents the dormant season nighttime; [6] DT represents the dormant season transition period.

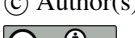



[7] MBF represents mixed broad-leaved forest; [8] MOF represents Mongolian oak forest; [9] LPF
represents Larch plantation forest.

## 4 Discussion

4.1 Short-term $[CO_2]$ fluctuations above the forest canopy and $F_s$ estimates in complex

terrains

This study found that short-term fluctuations of $[CO_2]$ above the canopy exhibited

a range of 1 to 10 min (Fig. 2). These fluctuations were characterized by an average

$P_{max}$ ranging from 2.313 to 2.784 min (Table 2). Our results are in line with previous

research using wavelet analysis, which reported fluctuation periods of $[CO_2]$ within and

above the forest canopy to be between 14 and 116 s (Cava et al., 2004). Their

observations of the canopy waves during periods of extreme atmospheric stability

(when $z/L \gg 1$) exhibited a dominant period of 1–2 min, consistent with our findings.

The period of $[CO_2]$ fluctuations was found to be predominantly influenced by turbulent

fluxes and the residence time of $CO_2$ within the canopy. This indicated a potential

correlation between $P_{max}$ and the residence time of $CO_2$ within the canopy. Fuentes et

al. (2006) employed a Lagrangian model and calculated the residence time of air parcels

released near the ground and canopy, finding values ranging from 3 to 10 min and from

1 to 10 min, respectively. Similarly, Edburg et al. (2011) used the standard deviation of

$[CO_2]$ averages to determine $CO_2$ residence time at different locations, including the

ground, within the canopy, and in their gas mixtures, yielding values of 8.6, 3.6, and

5.6 min, respectively. The results of these simulation experiments are obtained from our

study, further supporting the association between $[CO_2]$ fluctuations above the forest





canopy and $CO_2$ residence time.

Tree density and canopy structure also play a role in influencing the air parcel

residence time; in flat terrains, the air parcel residence time correlate with $u_*$ (Gerken
et al., 2017), and an increase in vegetation leaf area leads to longer residence times
when turbulence is not fully penetrative. During the growing season, forests exhibit
higher leaf area index and canopy densities compared to the dormant season, resulting
in longer $P_{max}$ of short-term $[CO_2]$ fluctuations above the canopy (Fig. 3). Additionally,
at night, stable atmospheric conditions lead to longer residence times due to suppressed
turbulent mixing, resulting in relatively long nighttime $P_{max}$ values compared to
daytime and transition periods (Fig. 3).

Complex terrains introduce multiple factors that influence $[CO_2]$ fluctuations,

including gravity-induced waves, drainage, and advection. These contribute to
uncertainties in estimating $F_s$. During nighttime, long-wave radiation emitted from the
valley soil surface leads to the cooling and downslope acceleration of air near the soil
surface due to gravity, potentially causing katabatic flow. As inertia-driven upslope
winds are halted by katabatic acceleration, a local shallow drainage flow is established,
reaching a quasi-equilibrium state approximately 1.5 h after sunset (Nadeau et al., 2013).
Under stable atmospheric conditions, even gentle slopes (around 1°) can generate
strong gravity-driven waves (Belušić and Mahrt, 2012). Consequently, advection may
complicate the interpretation of nighttime EC measurements at certain relatively gentle
sites, but this complexity is not evident during daytime measurements (Leuning et al.,
2008). Advection plays a role in depleting the $CO_2$ accumulated within the canopy,
resulting in lower $F_s$ fluxes and establishing an inverse relationship between storage
and advection (Van Gorsel et al., 2011). The occurrence of larger $F_s$ values for long $P_{max}$
values suggests weaker advection compared to short $P_{max}$ values (Fig. 4). In our study,
we observed that the $F_s$ magnitude was relatively large during nighttime and transition
periods, while it was smaller during daytime (Fig. 4), which is consistent with the
findings reported by Wang et al. (2016).
During nighttime and transition periods in a closed canopy, the turbulent coupling
state above and below the canopy gradually decouples, eventually reaching complete
decoupling as the $u_*$ decreases (Fig. 5). However, this decoupling does not lead to stable
stratification within the canopy. Despite the occurrence of decoupling and advection in
the closed canopy, waves are unlikely to exist within the canopy itself (Van Gorsel et
al., 2011). As a result, a consistent trend in the variation of $F_s$ with $\tau$ is observed across
the three forest stands during the growing season, independent of $P_{max}$ (Fig. 9).
Conversely, in an open canopy where waves are present, the observations of $F_s$ become
more complex. This complexity could be the primary reason why the variation of $F_s$
with [$CO_2$] averaging time windows differs between the three forest stands for short
$P_{max}$ values during the dormant season daytime (Fig. 9). The presence of waves
introduces additional variability in the measurements, leading to differences in $F_s$
estimates based on different [$CO_2$] averaging time windows in these particular
conditions.
4.2 Uncertainty in forest ecosystem $F_s$ measurement in complex terrains
Previous studies have highlighted the significant the random uncertainty of $F_s$ in



an open-canopy forest approximately 0.9 μmol m$^{-2}$ s$^{-1}$, compared to the measured
change in $F_s$ of 0.3 μmol m$^{-2}$ s$^{-1}$ and the estimated NEE of 6.0 μmol m$^{-2}$ s$^{-1}$ (Van Gorsel
et al., 2009). In the current study, we found that the uncertainty of $F_s$ estimates was
close to 2 μmol m$^{-2}$ s$^{-1}$ and 1 μmol m$^{-2}$ s$^{-1}$ for the growing and dormant seasons,
respectively, as $F_s$ approached zero (Fig. 7). The result for the dormant season was
consistent with the previous findings. However, the estimation method employed in our
study, comparing observations at two similar moments and ambient conditions, is
susceptible to environment changes and flux footprint variability, potentially leading to
an overestimation of the total random uncertainty in $F_s$.

The random uncertainty of $F_s$ shares similarities with NEE estimation. For

example, the magnitude of $F_s$ measurements is positively correlated with the standard
deviation of random uncertainty in $F_s$. Additionally, the overall distribution of $F_s$
measurements exhibits a non-Gaussian distribution with a high peak, aligning with the
statistical properties of NEE uncertainty (Richardson et al., 2006; Richardson et al.,
2008). Various factors contribute to the uncertainty in $F_s$ estimates, including flux
measurement footprint variations, sampling frequency, spatial sampling resolution of
$CO_2/H_2O$ concentrations, and instrumental measurement accuracy. The uncertainty
arising from variations in the flux measurement footprint is considerable, typically on
the order of tens of percentages, which is an order of magnitude higher than typical
sensor errors (Metzger, 2018). The AP200 atmospheric profiling system used in this
study has an accuracy of ±0.5 μmol mol$^{-1}$ and ±0.1 mmol mol$^{-1}$ for $CO_2$ and $H_2O$
concentration measurements, respectively (Montagnani et al., 2018). Efforts to reduce

598 random errors in [$CO_2$] originating from pressure fluctuations include adding buffer

599 volumes before IRGA pumping tests (Marcolla et al., 2014). The AP200 adopts buffer

600 volumes that are fully mixed during gas extraction and performs a weighted average of

601 [$CO_2$] instantaneous measurements to minimize the sampling error for each level's

602 [$CO_2$] measurement (Cescatti et al., 2016).

603   The $F_s$ estimates can be influenced by singular eddies that penetrate inside the

604 canopy (Finnigan, 2006). Accurate calculation of $F_s$ requires considering the period of

605 [$CO_2$] fluctuations with the eddy coherence structure. The spectral energy of the $F_s$ time

606 series is primarily concentrated between 0.001 and 0.2 Hz (500 and 5 s, respectively).

607 However, even with sampling frequencies of 2 Hz and below, significantly lower $F_s$

608 values are obtained (Bjorkegren et al., 2015). The Nyquist-Shannon sampling theorem

609 dictates that accurate measurements of [$CO_2$] require a sampling period no longer than

610 half the period of [$CO_2$] fluctuations. Consequently, to monitor short-term changes in

611 [$CO_2$], measurements must be taken over a period no longer than half of the period

612 corresponding to the maximum amplitude (or major energy) of [$CO_2$] fluctuations. In

613 this study, the average $P_{max}$ for [$CO_2$] fluctuations fell within the range of 2.313–2.784

614 min (Table 2). Therefore, it is crucial to ensure that the sampling period for [$CO_2$] does

615 not exceed 1.256 to 1.392 min, which corresponds to half the average $P_{max}$ range.

616 Monitoring fluctuations of $P_{max}$ for less than 4 min during a 2-min monitoring period

617 of [$CO_2$] presents a significant challenge. This is a primary reason that the systematic

618 bias and random error in $F_s$ estimate with a single profile system are irreconcilable

619 (Wang et al., 2016). Short-term [$CO_2$] fluctuations are mainly influenced by boundary



layer turbulence, and sampling errors in incomplete fluctuation cycles will be
superimposed with the real advection flux (anisotropy) dispersion in complex terrains
(Van Gorsel et al., 2011). This substantially increases the random uncertainty in $F_s$
based on shorter [$CO_2$] averaging time windows (Fig. 6 and Fig. 8). As a result, the
deviation of NEE estimates from the actual value expands.
In complex terrains, the bidirectional airflow within forests along slopes can cause
the decoupling of soil $CO_2$ fluxes from EC measurements above the forest canopy
(Feigenwinter et al., 2008; Aubinet et al., 2003), leading to significant errors in $CO_2$
flux measurements. Forest soil serves as the primary source of $CO_2$ gas and regions of
high flux over complex terrains act like chimneys, transporting air parcels from the soil
surface within forests (Chen et al., 2019). In situations where turbulence is not well-
developed, and $CO_2$ mixing is inadequate, the trend of $F_s$ with turbulence intensity
aligns with that of advective fluxes, which is opposite to that of turbulent fluxes
(Mchugh et al., 2017). The temporal dynamics and amplitudes of $F_s$ changes are
influenced by topography complexity and wind conditions above the forest canopy (Fig.
10). Locations with more complex and sloping topography at the flux tower are more
likely to generate advective fluxes that may not be easily observed at a single point.
Estimating landscape $CO_2$ fluxes in complex terrains solely based on
measurements from a single flux tower can introduce significant errors and biases that
are not acceptable. The magnitude of these errors in $F_s$ estimates is dependent on the
height of the forest canopy and the endogenous source/sink (Chen et al., 2020). To
mitigate errors and biases associated with estimating $F_s$ in complex terrains, we



employed a regression modeling approach using the decision-level fusion model. This method involves computing a weighted average of $F_s$ based on different $[CO_2]$ averaging time windows, effectively reducing errors and biases in the estimation of $F_s$ (see Table 5). In fact, from the definition of storage flux, it can be seen that weighting the storage flux is essentially weighting the $[CO_2]$ in the average time window, which means replacing spatial sequences with temporal sequences for weighting. The weighting coefficients used to construct the model were based on the relative errors and biases of $F_s$ estimation, with the weighting coefficient decreasing as the represented moment's length increased. To obtain more accurate estimates of forest ecosystem $F_s$ in complex terrains, further research should focus on understanding the spatiotemporal patterns and dynamics of $[CO_2]$.

**5 Conclusions**

This study investigated the impact of short-term $[CO_2]$ fluctuations on the estimation of $F_s$ in temperate forest ecosystems within complex terrains. Additionally, it examined the $F_s$ uncertainty and the contribution of the $F_s$ to NEE using data from three flux towers. To enhance $F_s$ uncertainty estimation, statistical sampling techniques were applied based on the individual tower approach.

The results highlighted the significance of considering multiple time windows for averaging $[CO_2]$ when estimating $F_s$, as $[CO_2]$ above the forest canopies exhibited fluctuations with periods ranging from 1 to 10 minutes. Diurnal, seasonal, and spatial variations were observed in the amplitude and periodicity of $[CO_2]$ fluctuations,





highlighting the need for thoughtful sampling strategies. The use of individual gas
analyzers to sample the $CO_2$ in the control volume was inadequate, leading to
systematic biases and random errors in the $F_s$ estimates. Increasing $[CO_2]$ averaging
time windows mitigated the effect of $[CO_2]$ fluctuations on $F_s$ estimates, reducing both
magnitude and uncertainty.

The study also revealed that the uncertainty of $F_s$ followed a non-normal

distribution, with its standard deviation positively correlated with $F_s$ magnitude, which
has important implications for quality control. To improve $F_s$ estimation, a decision-
level fusion model was introduced, integrating $F_s$ estimates from multiple $[CO_2]$
averaging time windows, effectively reducing the impact of short-term $[CO_2]$
fluctuations while considering underestimation bias and random errors. The
contribution of $F_s$ to NEE exhibited diurnal, seasonal, and spatial variations associated
with $u_*$, contributing to the NEE observations at rates ranging from 17.2% to 82.0%
depending on the turbulent mixing and terrain complexity. The influence of terrain
complexity on the relationship between $[CO_2]$ fluctuations, turbulent mixing, and the
contribution of $F_s$ to NEE was also evident. The findings from the three flux towers
allowed for the generalization of these results beyond the study site. These insights
provide crucial scientific support for the practical application of the eddy covariance
technique and advance our understanding of accurately estimating NEE in forest
ecosystems in complex terrains.

**Appendix A**





*A.1 the weight parameters of the decision-level fusion model*
For each 30-min $CO_2$ storage flux ($F_s$) estimate based on the $CO_2$ concentration
($[CO_2]$) averaging time window ($\tau$), the weight in the decision-level fusion model can
be obtained by weighting the random uncertainty and bias of $F_{s\_\tau}$.
The weight of the random uncertainty for the $F_{s\_\tau}$ is expressed as follows:

$$w_\tau = \frac{1/\sigma(\varepsilon_\tau)}{\sum_j 1/\sigma(\varepsilon_\tau)}, \tag{A.1}$$

where $\sigma(\varepsilon_\tau)$ is the random uncertainty of the $F_{s\_\tau}$, qualified as the standard deviation.
The weight of the bias for the $F_{s\_\tau}$ is expressed as follows:

$$W_\tau = \frac{K_\tau}{\sum_j K_j}, \tag{A.2}$$

where $K_\tau$ is the slope between the $F_{s\_\tau}$ and $F_{s\_28}$.
Ultimately, the weight of the $F_{s\_\tau}$ in the decision-level fusion model can be
calculated using the following equation:

$$w_\tau^* = rw_\tau + (1-r)W_\tau, \tag{A.3}$$

where $r$ represents the proportion of the weight of random uncertainty.
*A.2 Complex terrain index*
This study employed a novel descriptor called the terrain complexity index (*TCI*)
to quantify the complexity of the three-dimensional terrain. For a given unit area, the
*TCI* equation can be expressed as follows:

$$TCI = (1 - P_d\cos\alpha)(1 - Z_d^{-1})(D_f - 2)^{H/\ln(12)}, \tag{A.4}$$

where, $P_d$ represented the volume of terrain above the lowest elevation of an area unit
($V_u$) divided by the product of its largest vertically projected area ($S_v$) and the edge
length of the side of the area unit ($d$), expressed as $P_d = V_u/(S_v d)$; $P_d$ was defined to
be one when the $S_v$ is zero. $\alpha_d$ indicated the slope of the area unit. $Z_d$ denoted the
terrain roughness, which defined as the ratio of the terrain surface area to the projected
horizontal plane. $D_f$ is the fractal dimension of terrain surface area, which ranged from
2 to 3 and described the complexity in spatially self-similar structure of the local surface
within the area unit and the area unit surface (B. B. Mandelbrot, 1967; Taud and Parrot,
2005). Employing terrain surface area, the box-counting method is used to estimate
fractal dimension of unit area. $H$ denoted the Shannon-Wiener index and expressed as
$H = -\sum_{i=1}^{n} P_i \ln(P_i)$, capturing the uniformity of the spatial distribution of the pixel
aspects within the area unit. When the aspect of each pixel was divided into 30°
segments, $P_i$ denoted the proportion of the $i^{th}$ type of pixel aspects within the area unit
and $n$ was the total number of pixel aspect types within the area unit.
To quantify the terrain complexity of the underlying surface around the flux towers,
we computed the quartiles of *TCI* for all area units within a sector (divided by 30°) with
a radius of 380 m. A weighted geometric mean was employed to construct *TCIs*, which
describe the statistical distribution of *TCI* of the sector. The *TCIs* represents the
topographic complexity of the sector and are calculated using the following equation:
$$TCI_s = (TCI_5 TCI_{25} TCI_{50} TCI_{75} TCI_{95})^{1/5} \tag{A.5}$$
where *TCI*₅, *TCI*₂₅, *TCI*₅₀, *TCI*₇₅, and *TCI*₉₅ are the quartiles of 5%, 25%, 50%, 75%,
and 95%, respectively. The *TCIs* values range from 0 to 1, with higher values indicating
greater terrain complexity.
*Data availability.* Data used in this paper are available at the Science Data Bank



(https://www.scidb.cn/en/s/7ZfQZv) or upon request to the corresponding author.
*Author contributions.* DT developed the manuscript; JZ was responsible for
conceptualizing the idea and designing the research study; TG substantially structured
the manuscript; FY contributed to the data collection process; YZ helped in the design
and preparation of the figures and tables; XZ and BY revised the manuscript.
*Competing interests.* The authors declare that they have no known competing
financial interests or personal relationships that could have appeared to influence the
work reported in this paper.
*Acknowledgments.* We are grateful to Qingyuan Forest CERN, Chinese Academy of
Sciences/Qingyuan Forest, National Observation and Research Station, Liaoning
Province, China for providing forest sites, instrument systems, and logistic supports.
*Financial support.* This research was financially supported by the National Natural
Science Foundation of China (No. 32192435), the China Postdoctoral Science
Foundation (No. 2023M733672), and the Postdoctoral Research Startup Foundation
of Liaoning Province of China (No. 2022-BS-022).

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
