# Peer review of "Field assessments on impact of CO2 concentration fluctuations along with complex"

_Atmospheric Measurement Techniques, 2024_

## Author Comment (AC1)

[Figure]

Figure S1 Response of daytime NEE observations (NEE$_{obs}$) to photosynthetic photon flux density (PPFD) during July and August

[Figure]

Figure S2 Response of nighttime NEE observations (NEE_obs) to air temperature (Ta) during the dormant season

The influences of $F_s$ on the relationship between NEE observations and meteorological drivers, indicated the effect of uncertainty in $F_s$ estimates on NEE observations. Our analysis showed that the correlations between NEE observations

derived from $F_c+F_s$ and both photosynthetic photon flux density (PPFD) and air temperature are lower compared to those obtained from $F_c$ alone (Figure 1 and Figure 2 in the Supplementary Materials). Additionally, the estimated light saturated net $CO_2$ assimilation ($A_{max}$) is greater when NEE observations are estimated by $F_s+F_c$, as opposed to when NEE is estimated solely by $F_c$. This suggests that $F_s$ significantly affects daytime NEE and can correct the estimation of $A_{max}$ and related parameters. The relationship between NEE observations and PPFD is influenced by the size of averaging time window the $F_s$ measurement. A larger averaging window results in less random uncertainty in the $F_s$ estimation, thereby increasing the correlation between NEE observations and meteorological drivers, including PPFD and Ta.

---

## Author Comment (AC2)

[Figure]

Fig. A3 Calculation the volume of terrain above the lowest elevation of an area unit ($V_u$) and its largest vertically projected area ($S_v$) utilizing 3-dimentional box counting. $d$ is the edge length of the side of the area unit; $V_u$ represents the cumulative volumes of the constituent cubes; and $S_v$ indicates the total area of the shaded regions. $P_d$ is calculated by the ratio of $V_u$ to the product of $S_v$ and $d$.

---

## Author Response (AR1)

Thank you so much for your favorable consideration to encourage us to revise our manuscript. Many thanks to Dr. Montagnani and the anonymous reviewer for their valuable input on our manuscript. Referees provided technical comments on computations as well as an insight into our study implication in the major comments. These comments and suggestions helped us clarify our own concerns and improve our manuscript.

As referees did not number their general comments, we have categorized the key feedback from the referees into seven main comments. Every major comment was discussed in details, and we revised the manuscript in response to all the comments and suggestions carefully. Almost all of the suggestions were accepted in the revised MS. The responses are given right below the comments or suggestions. The revised portions have been highlighted in red in the revised manuscript. The detailed responses are as follows (RMC: reviewer's major comments; RSC: reviewer's specific comments; AR: author's response).

**Referee 1**

**General comments:**

**RMC 1.1**: The authors state that 'For 30-min and annual forest ecosystem carbon flux measurements, ignoring Fs would underestimate the NEE (Zhang et al., 2010)'. This idea is then repeated by Li et al., 2020, and the authors based on a very complex and uncommon computation support this idea. I am sceptical about these findings. To my knowledge, the storage flux has a relevant impact on daily NEE course, but no or negligible impact on cumulative NEE. I believe the authors have made a mistake in the computation, or possibly it is a result of a wrong gap-filling procedure.

**AR**: Theoretically, the impact of storage flux on the annual cumulative NEE can be neglected, but the storage flux has a relevant impact on half-hourly NEE course. For example, a significant $CO_2$ emission in the forest ecosystem is usually observed in the early mornings as the accumulation of $CO_2$ overnight released instantly. This "flush" has been repeatedly reported in field (Novick et al., 2014). Neglect of $CO_2$ storage could indicate that the ecosystem is a $CO_2$ source, leading to mistaken estimation of half-hourly NEE. In addition, our findings showed that the contribution of storage

flux to half-hourly NEE observations is significant even under the condition of strong turbulence. In practice, it is necessary to fill the gaps in half-hourly NEE time series when calculate the annual cumulative NEE. Since gap filling relies on the half-hourly NEE data, the mistaken estimation of half-hourly NEE may propagate to the cumulative NEE.

The mentioned statement in the introduction section aims to highlight the significance of half-hourly $CO_2$ storage. Our study did not perform any gap filling for $CO_2$ concentration, storage flux, and NEE observations, and all the analysis was based on observation data. Instead, Zhang et al., 2010 and Li et al., 2020 did gap-filling procedure in their works. To express clearly, we rewrite these sentences (see Lines 70-74 and Lines 107-108, respectively) as "For 30-min ecosystem carbon flux measurements, ignoring $F_s$ would underestimate the NEE (Zhang et al., 2010). The $F_s$ value typically ranges from −2 to −5 μmol m$^{-2}$ s$^{-1}$ in the early morning, and the $F_s$ is about 1–3 μmol m$^{-2}$ s$^{-1}$ after sunset for temperate forests. The effect of the $F_s$ on the NEE of forest ecosystems decreases with the increase of timescale (Li et al., 2020)." and "A proper measuring system with improving the horizontal representativeness can reduce the bias of $F_s$ to 2–10% (Nicolini et al., 2018)."

**RMC 1.2**: Although my opinion is possibly biased by my involvement in the ICOS network, I cannot understand why the authors did not consider the instrumental setup and the computational procedure we developed and applied in Europe. At least in theory, the use of spatially distributed air intakes in the control volume would overcome the problems arising from the presence of single gusts of CO2-rich air.

**AR**: We have studied the guidelines of ICOS Ecosystem Instructions that provide a technical solution to minimize the problems of inadequate spatial sampling of gas concentration from a single intake at one height. We have used AP200 Atmospheric Systems (Campbell Scientific Inc., UT, USA) for $CO_2$ and $H_2O$ profiles. In this system, air is mixed at each level inside a mixing bottle, which reduces the error due to the fluctuations from a single gust. Modification of AP200 to follow ICOS protocol is not an easy engineering task, without manufacturer's help, beyond our ability. Additionally, we had to follow the conventions of using AP200 series in ChinaFlux network, in which the manufacturer's guides are used in the installation for long-term measurements. The computation procedure embedded into the AP200 code, which has

been adopted for raw data collections.

We have realized that we need to reconcile the AP200 configurations and algorithms with the ICOS setting and computation procedure in our future applications of AP200 or other Atmospheric Profile Systems if any.

To statement comprehensively, we have discussed ICOS and spatial sampling in discussion. We have added the sentence in discussion section as follow (see Lines 650-652): "By increasing the number of gas concentration sampling points near the ground, the horizontal representativeness can be enhanced, thereby reducing the bias in the estimation of $F_s$ (Nicolini et al., 2018)."

**RMC 1.3**: Within ICOS, the computational period is defined between the concentration measurements done around the beginning and the end of the half-hour, while the authors consider a large interval, in my view possibly not compliant with the Reynolds decomposition.

**AR**: We agree with the reviewer on this point. Actually, the computational period for estimating storage flux is the same as averaging period of turbulent flux, which is half-hour in this study. However, using a single gas analyzer to measure $CO_2$ density at multiple levels, the synchronous observation of a $CO_2$ profile is practically impossible. Sampling at one level takes about 15 s (with 10 s for the flushing of the manifold and 5 s for logging the average), leading to a measurement cycle of 2 min for 8 levels in total. In addition, discrete time averaging is usually needed to remove the noise in the time series of $CO_2$ records. With this consideration, during routine operation at our site we compute the storage flux term using the $CO_2$ values within a 2-min window (the shortest possible) around the beginning and ending of each half-hour interval. This approach is fully compliant to the principle of Reynolds decomposition.

In this study, we did expand the averaging windows to 4, 8, …, 28 min, respectively. The purpose was to demonstrate how enlarged windows around the beginning and ending of each half-hour interval for $CO_2$ averaging would affect the accuracy in storage flux estimates. We have rewritten these sentences into (see Lines 193-203): "When measuring the $F_s$ by sampling $CO_2$ at several levels using a single analyzer, the synchronous observations of $CO_2$ profile are impractical. Consequently, discrete

temporal sampling and time averaging become necessary. To ensure the temporal alignment of $F_s$ with $F_c$, the average [$CO_2$] measurements within the control volume at the beginning and end ($t$) of an averaging period (30 min) are calculated by averaging over a time window ($\tau$ min) as follows:

$$\bar{\chi}_{c_i} = \frac{2}{\tau} \sum_{t-\frac{\tau}{2}<t\leq t+\frac{\tau}{2}} \chi_{c_i}(t), \tag{7}$$

where $\tau = 4, 8, \ldots, 28$ min. Theoretically, the time window should be kept as short as possible in comparison to the turbulence flux averaging period to comply with the principle of Reynolds decomposition. We use large windows here for $CO_2$ averaging in an attempt to demonstrate the effects of different window sizes on the accuracy of storage flux estimates."

**RMC 1.4**: I would also like to see what the effect of the CO2 storage flux is when combined with the turbulent component, if it improves the correlation with environmental drivers, like PPFD, or if it deteriorates it.

**AR**: We appreciate your thoughtful comment. The analysis provided indeed underscores the significance of $F_s$. As you suggested, we examined the correlation between NEE observations and PPFD as well as air temperature. The figures reveal that the combination of Fs and Fc exhibits a superior fit with PPFD compared to $F_c$ alone, while it demonstrates a comparatively less favorable fit with air temperature (Ta) than $F_c$ alone. In response to this observation, we have incorporated two supplementary figures to visually depict the corrected correlation with these environmental drivers when $CO_2$ storage is taken into account. These figures are uploaded as supplementary material (Figure S1 and S2).

**Special comments:**

**RSC 1.1**: Line (L) 30. $A_{max}$ is an acronym largely used to represent maximum assimilation capacity through photosynthesis. I find it confusing to use the same acronym for another meaning.

**AR**: We have modified $A_{max}$ as $A_m$. Correspondingly, we also modified $P_{max}$ to $P_m$.

**RSC 1.2**: L86: '…cold air moves from the ground to the valley forest canopy'. In principle, cold air tends to move downward since it is heavier, so it does not reach the canopy. This process is the basis of Katabatic flows and advection. Please explain.

**AR**: Thank you for pointing this mistake. We modified this sentence as "…cold air moves from the valley forest canopy to the ground".

**RSC 1.3**: L93: 'In practice, the Fs represents the integration of the time derivative of the vertically determined column-averaged [$CO_2$].' I believe that this 'practical' and traditional assumption is wrong and undermines the possibility of correctly measuring the storage flux since a profiling system is not representative of the whole control volume. The authors cite several times the paper by Nicolini et al. but they do not take advantage of that study.

**AR**: Thanks for your comments. In practical, the NEE observations were obtained through the jointly measurements of $F_s$ and $F_c$. Under the conditions of well-developed turbulence or well-mixed air, these "practical" and traditional assumptions are correct. This implies that an accurate estimation of $F_s$ can be obtained using a single gas analyzer in a relatively economical manner. When these conditions are not met, as you said, these assumptions do not hold. We rewrite these sentence as (see Lines 96-101) "The estimation of Fs at numerous sites frequently employs a vertical profile system. This is based on the assumption that the $F_s$ represents the integration of the time derivative of the vertically determined column-averaged [$CO_2$]. However, the column-averaged [$CO_2$] cannot well represent the average [$CO_2$] of the control volume due to insufficient sampling when the air mixing is inadequate."

**RSC 1.4**: L126: 'lack of clear guidelines'. Being part of ICOS, I disagree with this statement. We wrote a protocol, a paper and the instructions are publicly available, https://fileshare.icos-cp.eu/s/LzAZfgkc4znRtz6

**AR**: Thanks for your suggestions. We acknowledge that this ICOS Ecosystem Instructions provide good guidelines for estimating storage flux. This sentence was modified as (see Lines 131-133) "Furthermore, resource constraints in the measurement system leads to the gap that the systematic bias and random error in $F_s$ estimate are irreconcilable."

**RSC 1.5**: L185. I did not check all the equations reported, but to my knowledge, the $CO_2$ molar mass is around 44 g mol-1, and not 12.011 g mol -1. This is the mass of the carbon only.

**AR**: Thank you for pointing this mistake. We thoroughly checked all the equations. The eq.5 was accepted in CIPM (Picard et al., 2008), where Mc is the carbon molar mass (12.011 g mol$^{-1}$). We have corrected this mistake in the revised MS.

**RSC 1.6**: L242. 'The average Fs was calculated in a certain time window (15 d)…'. Why do the authors make such a computation? Probably the result of a significant value of Fs in the long term comes from this averaging. Did the authors perform a gap-filling, quality test or other? And how did they fill the gaps, if any? These are delicate passages in which a wrong computational window or filling procedure can have an impact.

**AR**: The goal of this computation is to estimate the uncertainty of the storage flux, rather than estimating the $F_s$ values. We rewrite the steps of this computations and add five formulas in the section 2.4.

As mentioned above (**RMC 1**), we did not perform any gap filling. Indeed, we only removed extreme outliers (the standard is $4\sigma$) in $CO_2$ concentration, storage flux, and turbulence flux time series.

*Proposed revision* (see Lines 254-263):

To determine the uncertainty of $F_s$, expressed as $\sigma(\varepsilon_s)$, this study compared the observations at moment *i* to the average of several observations during a similar period and with similar meteorological conditions. The specific computations were as follows:

$$\overline{F}_s^{(i)} = \frac{1}{N} \sum_{t\in\Omega, \lambda_t\in\Lambda} I(\lambda_t) \cdot F_s^{(t)}, \tag{11}$$

$$\Lambda = \{\lambda_t | \sqrt{\frac{\left(u_*^{(\lambda_t)} - u_*^{(i)}\right)^2}{\sigma_{u_*}} + \frac{(Ta^{(\lambda_t)} - Ta^{(i)})^2}{\sigma_{Ta}} + \frac{(H^{(\lambda_t)} - H^{(i)})^2}{\sigma_H}} < \delta\}, \tag{12}$$

$$\varepsilon_s^{(i)} = F_s^{(i)} - \overline{F}_s^{(i)}, \tag{13}$$

$$\bar{\varepsilon}_s^{(i)} = \frac{1}{N} \sum_{t \in \Omega, \lambda_t \in \Lambda} I(\lambda_t) \cdot \varepsilon_s^{(t)}, \tag{14}$$

$$\sigma(\varepsilon_s)^{(i)} = \sqrt{\frac{1}{N} \sum_{t \in \Omega, \lambda_t \in \Lambda} I(\lambda_t) \cdot (\varepsilon_s^{(t)} - \bar{\varepsilon}_s^{(i)})^2}, \tag{15}$$

where $\Omega$ was the moment interval ($i-0.5$ h, $i+0.5$ h) within a certain time window (15 d); $I$ was indicator function; the set $\Lambda$ represented consisted of elements that meet similar meteorological conditions, including the $u_*$, air temperature (Ta), and sensible heat flux (H); $\sigma_{u_*}$, $\sigma_{\text{Ta}}$, and $\sigma_H$ are the standard deviation of the $u_*$, Ta, and H, respectively; $\delta$ was the threshold of Euclidean distance; and $\varepsilon_s$ was the random error of $F_s$.

**RSC 1.7**: I find Figure 12 very interesting, well done. I must be sincere, I cannot understand what the authors try to explain in line 485 and the following.

**AR**: The goal of line 485 and the following was to evaluate the difference in $F_s$ estimates between the dynamic-time-window and the fixed-time-window method. For the practical application, we also assessed the role of our efforts in estimating $F_s$ on the observations of NEE. Although it is very difficult to completely eliminate the uncertainties in NEE observations, the analytical results of the MS indicate that our efforts reveal statistically significant meaning.

**RSC 1.8**: I believe that Tables 4 and 5 contain too many values to be informative, better placing them as supporting material or making a synthesis.

**AR**: Thank you for your suggestion. Tables 4 and 5 have been removed. We placed them as supporting material.

**RSC 1.9**: L542: 'Complex terrains introduce multiple factors that influence [CO2] fluctuations, including gravity-induced waves, drainage, and advection. These contribute to uncertainties in estimating Fs.'. I am not fully convinced. Advection is a separate term from storage, and storage cannot account for advection. It is a different term.

**AR**: Thanks so much for reviewer to provide this theoretical insight. The uncertainties in estimating $F_s$ depends on the vertical configuration of the profiling system and the

number of sampling levels (Yang et al., 2007). The optimal vertical configuration should be subject to the structures of air flow, which predominantly governs the $[CO_2]$ fluctuations. Therefore, the systematic bias and random error in $F_s$ estimate with a single profile system are irreconcilable. Acknowledging this point, we rewrite these sentences (see Lines 560-563 in the Section 4.1) as "Complex terrains introduce complex changes in air flow structures, including gravity-induced waves, drainage, and nonlinear waves induced by single gusts, leading to dramatic $[CO_2]$ fluctuations. These dynamics contribute to uncertainties in estimating $F_s$."

**RSC 1.10**: L574 and following. Again, the uncertainty in the storage term depends a lot on the set-up used, together with the biological activity of the ecosystem and the height of the control volume. It does not make much sense to discuss typical uncertainty.

**AR**: We agree with your understanding of the uncertainty in the storage term. We delete L574 and following. Indeed, the typical uncertainty as $F_s$ approached zero might be site specific.

**RSC 1.11**: L597: In Montagnani et al., 2018 it was not discussed the AP200.

**AR**: Thank you for pointing this mistake. We had modified the reference as (Cescatti et al., 2016) in the revised MS. We have rewritten this sentence into (see Lines 614-616 in the Section 4.2) "The buffer volumes are fully mixed during gas extraction and performs a weighted average of $[CO_2]$ instantaneous measurements to minimize the sampling error for each level's $[CO_2]$ measurement (Cescatti et al., 2016)."

**Referee 2**

**General comments**:

**RMC 2.1**: I think the title of the work is very striking, but it seems to me that the discussion about the complexity of the terrain went to the background, the discussion about this issue should be broader.

**AR**: Thanks for the comment. In the discussion section, we have broadened the discussion to incorporate a comprehensive analysis of $CO_2$ concentration variability and the methods employed for flux measurement within complex terrain.

*Proposed revision*:

Compared to flat and uniform underlying surface, complex terrain and heterogeneous canopies modify the trajectory, speed distribution and direction of the airflow. Increased wind speeds and shifting wind directions also increase turbulent activity above the canopy, facilitating the mixing and dispersion of $CO_2$. (Lines 529-532 in the Section 4.1)

The terrain complexity and the diversity within the canopy significantly affect the airflow separation in the atmospheric boundary layer. This results in weakened air circulation within the canopy and spatial variation in the patterns and extent of airflow separation (Grant et al., 2015). (Lines 579-582 in Section 4.1)

Fluxes are significantly higher in heterogeneous regions than in uniform regions. The energy transfer from the ground surface to large eddies occurs primarily in areas with pronounced heterogeneity, and this energy distribution is uneven across the region (Aubinet et al., 2012). Once large-scale eddies acquire energy, their cascading of energy to smaller-scale eddies is influenced by topographic features, leading to variations in these smaller-scale eddies along different flow streams (Chen et al., 2023). (Lines 639-644 in Section 4.2)

**RMC 2.2**: I think that the Appendix about the TCI needs at least a couple of references and more context (images), since it is a complex topic to understand and it is not clear if the authors propose the descriptor or it was previously established.

**AR**: Thanks for the comment. The terrain's features and spatial element distribution were considered when developing the TCI. Additionally, the relationship between the local and the whole was taken into account. Quantifying terrain complexity is not the focus of this MS, therefore it will not be discussed in depth. We are currently working on a MS that aims to quantify terrain complexity.

We have created a conceptual diagram to illustrate the quantitative assessment of volume filling ratio ($P_d$) in details (Figure A3). This figure provides the estimations of the volume of terrain above the relatively lowest elevation of an area unit ($V_u$) and its largest vertically projected area ($S_v$). Therefore, the quantitative estimation of $P_d$ is methodically derived.

We have included corresponding references for roughness and the Shannon Wiener index. To provide a clearer introduction to the TCI descriptor, we have expanded the details in Appendix A.2.

*Proposed revision:*

Given $V_u$, an increase in $S_v$ correlates with a higher degree of terrain complexity. Notably, the $P_d$ is defined as 1 when the terrain volume is 0 or when the terrain surface of the area unit was parallel to the horizontal plane and was smooth and homogeneous. (Lines 726-728)

The value of $Z_d$ is in the range of [1, +∞). The larger $Z_d$, the more complex the terrain. (Lines 730-731)

A larger $H$ indicates a more complex terrain. When the number of pixel aspect types in the area unit is kept constant, it's essential to recognize that greater uniformity in the distribution of all pixel aspect in the area unit results in a larger $H$. Similarly, when the uniformity of the distribution of pixel aspects in the area unit is kept constant, a larger $H$ is achieved with an increase in the observation of the number of pixel aspect types. (Lines 741-745)

**RMC 2.3**: A couple of figures in the paper (Figs. 10 and 11) need some quality touches, since it is not possible to clearly see the legend to distinguish the lines.

**AR**: Thanks so much for this suggestion. We have redrawn the Figures 10 and 11 in bitmap format.

**Special comments**:

**RSC 2.1**: L. 97: Personally, I think you should use numbers when referring to time averages along all the text. So replace two-min by 2-min

**AR**: Revised.

**RSC 2.2**: L. 158: The exact name of integrated system from Campbell Scientific you are using is the CPEC310. So for me it would be better read something like: "The CPEC310 integrated system from Campbell Scientific comprising an ..."

**AR**: Thanks for the suggestion. We have rewritten this sentence (Lines 163-165 in the Section 2.1) as "The CPEC310 integrated system from Campbell Scientific comprising an EC155 closed-path infrared gas analyzer (IRGA) and a CSAT3A sonic anemometer, was employed to monitor the three-dimensional wind speed and $CO_2/H_2O$ concentrations (10 Hz)".

**RSC 2.3**: L. 159: Remove "ray" word. IRGA = InfraRed Gas Analyzer

**AR**: Thanks for the suggestion. It has been removed in the revised MS.

**RSC 2.4**: L. 175: replace "Dolton's" by "Dalton's"

**AR**: Replaced "Dolton's" by "Dalton's" in the revised MS.

**RSC 2.5**: L. 185: the $CO_2$ molar mass is 44.01 grams per mole.

**AR**: Thank you for pointing this mistake. As **RSC 1.5**, the eq.5 was accepted in CIPM (Picard et al., 2008), where $M_c$ is the carbon molar mass (12.011 g mol$^{-1}$). We corrected this mistake in the revised MS.

**RSC 2.6**: L. 218: Mention explicitly the average time windows you employed.

**AR**: We have included specific details regarding the averaging time windows in Lines 231-232.

*Proposed revision:*

CO$_2$ storage fluxes were calculated for different [CO$_2$] averaging time windows (τ), ranging from 4 to 28 min in increments of 4 min.

**RSC 2.7**: L. 225: In Equation (9) explain what the "i" index refers to

**AR**: Thanks for the comment. We calculated the normalized root mean square error to evaluate the relative error between F$_{s\_τ}$ and F$_{s\_28}$ using Equation (9). In this equation, *i* indicates the *i*[th] observation of F$_{s\_τ}$ or F$_{s\_28}$. This information has been added in the revised MS (Line 240)

**RSC 2.8**: L. 242-244: Why did you use 15d window? What does "*i* was located" mean?

**AR**: We employ multiple observations under similar meteorological conditions over a 15-d window to substitute for repeated observations. The meteorological conditions include friction velocity, air temperature, and sensible heat flux. Considering that the estimation of the sensitivity of flux measurements to temperature mostly adopts a 15-d window (Reichstein et al., 2005), this study also uses a 15-d window to find flux observations under similar meteorological conditions. Actually, the key constraint are the similar environmental conditions and the corresponding half-hour flux measurements at the time of a day rather than the window size (e.x., 15-d window). Here, '*i*' represents a specific half-hour period within a day.

To express clearly, we rewritten this sentence (Lines 254-256 in the Section 2.4) as "To determine the uncertainty of F$_s$, expressed as $\sigma(\varepsilon_s)$, in this case, we compared the observations at moment *i* within a day to the average of several observations during a similar period and with similar meteorological conditions."

**RSC 2.9**: L. 254: As mentioned previously: It seems not to be very comprehensible. More detail and references need to be added to the Appendix A.2.

**AR**: Please see our response to RMC 2.2.

**RSC 2.10**: L. 303-305: I am not sure if "amplitude" is the appropriate word, because you are referring to the "variation or magnitude" in the diurnal cycle. It might be confused with the amplitudes you got using EMD and spectral analysis.

**AR**: Thank you for your insightful comment. We appreciate your attention to this detail and ensure that the terminology is clear. When referring to the "variation or magnitude" in the diurnal cycle, our intention was to highlight the extent of change within this diurnal cycle. While "amplitude" is indeed commonly associated with the peak values in EMD and spectral analysis, in this specific context, it is used to describe the overall size or intensity of the short-term fluctuations of the $CO_2$ concentration.

To avoid any ambiguity, we have revised the terminology. Using terms like "extent of variation" in the diurnal cycle would be more appropriate and less likely to cause confusion with the amplitudes derived from EMD and spectral analysis.

**RSC 2.11**: L. 314: Caption in Fig. 5 replace "donate" by "indicate", or "represent"

**AR**: Replaced.

**RSC 2.12**: L. 355-357: I think it is very important to why the reduction in random error approaches to behavior of white noise.

**AR**: We appreciate your thoughtful comment. The random uncertainty of $F_s$ ($\sigma(\varepsilon_s)$) is positively correlated with the $F_s$ magnitude ($|F_s|$). This implies that $\sigma(\varepsilon_s)$ increases as the $|F_s|$ increases. The distribution of random error in $F_s$ can be regarded as a mixed Gaussian distribution with non-constant variance. Increasing the $[CO_2]$ averaging time window, results in a reduction of the random error in $F_s$ and the correlation coefficient between $\sigma(\varepsilon_s)$ and $|F_s|$. When the correlation coefficient is smaller and closer to zero, it indicates that the variation in $\sigma(\varepsilon_s)$ is smaller. In this case, the random error in $F_s$ is closer to white noise. To express clearly, we have rewritten the sentence (Lines 366-369 in the Section 3.2) to "These findings suggested that increasing the $[CO_2]$ averaging time window results in a reduction of the random error in $F_s$ and the correlation coefficient between $\sigma(\varepsilon_s)$ and $|F_s|$. This indicates a decrease in variability and a behavior similar to white noise."

**RSC 2.13**: L. 408: It is necessary to explain what is shown in Figure 10. Explain clearly each one of the predictors (independent variables) you employed in the multiple linear regression. Why does mean ln(A_max) and ln(P_max)? You can explain this better in the caption of the figure.

**AR**: Thanks for the comment. The use of logarithmic transformation on $[CO_2]$ fluctuations reduces data dispersion and helps to meet the assumption of homoscedasticity in regression analysis. We have revised the caption of the Figure 10 as follows: "Linear regression coefficients of the $CO_2$ storage flux ($F_s$) magnitude—driving factors relationships for the seven $CO_2$ concentration ($[CO_2]$) averaging time windows. The predictors of the multiple linear models are (a) the logarithm of maximum amplitude of $[CO_2]$ fluctuations ($\ln(A_m)$); (b) the logarithm of the corresponding period of maximum amplitude ($\ln(P_m)$); (c) the terrain complexity index (TCI); (d) the friction velocity ($u_*$); and (e) the interaction term of TCI and $u_*$, respectively. (f) $\beta_0$ represents the intercept term."

**RSC 2.14**: L. 419-422: Figures 10 and 11. Use the tags ((a), (b), (c), ...) from the plot to better describe the caption of the figure.

**AR**: We have rewritten the caption of Figures 10 and 11.

**RSC 2.15**: L. 428-442: This discussion is not easy to follow.

**AR**: Thanks for the comment. For greater clarity, we have added the following statements in the paragraph, respectively.

*Proposed revision:*

A multiple linear regression model was used to analyze the effect of $[CO_2]$ fluctuations on the random uncertainty of $F_s$, $\sigma(\varepsilon_s)$, in complex terrains. This model considered the interaction effects of $[CO_2]$ fluctuations and terrain complexity on $\sigma(\varepsilon_s)$, as shown in Fig. 11. (Lines 432-435)

The magnitude of these correlation coefficients decreased with the increasing $[CO_2]$ averaging time windows. (Lines 438-440)

These observations suggested that the relationship between the random uncertainty in $F_s$ and $[CO_2]$ fluctuations was moderated by topographic complexity. Increasing the

[CO₂] averaging time window reduced the effect of [CO₂] fluctuations on the random uncertainty in $F_s$. (449-452)

**RSC 2.16**: A.1 L. 689: Equation A.1. In the denominator is missing the "j" index

**AR**: Added the "*j*" index in Equation A.1.

**RSC 2.17**: A.2 L. A figure would be illustrative to understand what are P_d, S_v, etc.

**AR**: Thanks for the comment. We have added a figure (Figure A3) to explain the P_{d}, S_{u}, and S_{v}. This figure has been uploaded as supplementary material.

**Reference**

Picard, A., Davis, R. S., Gläser, M., and Fujii, K.: Revised formula for the density of moist air (CIPM-2007), Metrologia, 45, 149-155, 10.1088/0026-1394/45/2/004, 2008.

Reichstein, M., Falge, E., Baldocchi, D., Papale, D., Aubinet, M., Berbigier, P., Bernhofer, C., Buchmann, N., Gilmanov, T., Granier, A., Grunwald, T., Havrankova, K., Ilvesniemi, H., Janous, D., Knohl, A., Laurila, T., Lohila, A., Loustau, D., Matteucci, G., Meyers, T., Miglietta, F., Ourcival, J.-M., Pumpanen, J., Rambal, S., Rotenberg, E., Sanz, M., Tenhunen, J., Seufert, G., Vaccari, F., Vesala, T., Yakir, D., and Valentini, R.: On the separation of net ecosystem exchange into assimilation and ecosystem respiration: review and improved algorithm, Global Change Biology, 11, 1424-1439, 10.1111/j.1365-2486.2005.001002.x, 2005.

Aubinet, M., Vesala, T., and Papale, D.: Eddy Covariance: A Practical Guide to Measurement and Data Analysis, Springer Atmospheric Sciences, Springer, Dordrecht, XXII, 438 pp., 10.1007/978-94-007-2351-1, 2012.

Cescatti, A., Marcolla, B., Goded, I., and Gruening, C.: Optimal use of buffer volumes for the measurement of atmospheric gas concentration in multi-point systems, Atmospheric Measurement Techniques, 9, 4665-4672, 10.5194/amt-9-4665-2016, 2016.

Chen, J., Chen, X., Jia, W., Yu, Y., and Zhao, S.: Multi-sites observation of large-scale eddy in surface layer of Loess Plateau, Science China Earth Sciences, 66, 871–881, https://doi.org/10.1007/s11430-022-1035-4, 2023.

Grant, E. R., Ross, A. N., Gardiner, B. A., and Mobbs, S. D.: Field Observations of Canopy Flows over Complex Terrain, Boundary-Layer Meteorology, 156, 231-251, 10.1007/s10546-015-0015-y, 2015.

Nicolini, G., Aubinet, M., Feigenwinter, C., Heinesch, B., Lindroth, A., Mamadou, O., Moderow, U., Molder, M., Montagnani, L., Rebmann, C., and Papale, D.: Impact of CO2 storage flux sampling uncertainty on net ecosystem exchange measured by eddy covariance, Agricultural and Forest Meteorology, 248, 228-239, 10.1016/j.agrformet.2017.09.025, 2018.

Novick, K., Brantley, S., Miniat, C. F., Walker, J., and Vose, J. M.: Inferring the contribution of advection to total ecosystem scalar fluxes over a tall forest in complex terrain, Agricultural and

Forest Meteorology, 185, 1-13, 10.1016/j.agrformet.2013.10.010, 2014.

Picard, A., Davis, R. S., Gläser, M., and Fujii, K.: Revised formula for the density of moist air (CIPM-2007), Metrologia, 45, 149-155, 10.1088/0026-1394/45/2/004, 2008.

Reichstein, M., Falge, E., Baldocchi, D., Papale, D., Aubinet, M., Berbigier, P., Bernhofer, C., Buchmann, N., Gilmanov, T., Granier, A., Grunwald, T., Havrankova, K., Ilvesniemi, H., Janous, D., Knohl, A., Laurila, T., Lohila, A., Loustau, D., Matteucci, G., Meyers, T., Miglietta, F., Ourcival, J.-M., Pumpanen, J., Rambal, S., Rotenberg, E., Sanz, M., Tenhunen, J., Seufert, G., Vaccari, F., Vesala, T., Yakir, D., and Valentini, R.: On the separation of net ecosystem exchange into assimilation and ecosystem respiration: review and improved algorithm, Global Change Biology, 11, 1424-1439, 10.1111/j.1365-2486.2005.001002.x, 2005.

Yang, B., Hanson, P. J., Riggs, J. S., Pallardy, S. G., Heuer, M., Hosman, K. P., Meyers, T. P., Wullschleger, S. D., and Gu, L.-H.: Biases of CO2 storage in eddy flux measurements in a forest pertinent to vertical configurations of a profile system and CO2 density averaging, Journal of Geophysical Research, 112, 10.1029/2006jd008243, 2007.

---

## Referee Report (RR1)

**Referee report for amt-2024-6 (Teng et al., Field assessments on impact of CO2 concentration fluctuations along with complex terrain flows on the estimation of the net ecosystem exchange of temperate forests)**

**General Comments:**

I think this is a very important and good study. It addresses the role of short-term fluctuations of CO2 on the estimation of storage term (Fs) in forest over complex terrain, using an innovative method (decision-level fusion model), which proves quite useful. By analyzing specific time series of three towers and classifying them using different drivers. They also estimated and compared the storage flux using a 10Hz eddy-covariance system and an atmospheric profiling system, additionally multiple statistical methods to analyse the uncertainty of Fs were employed.

This work shows a great effort to establish a methodology to reduce inaccuracies when estimating NEE, focusing on the storage term using different experimental sites and seasons. I think the title of the work is very striking, but it seems to me that the discussion about the complexity of the terrain went to the background, the discussion about this issue should be broader. I think that the Appendix about the TCI needs at least a couple of references and more context (images), since it is a complex topic to understand and it is not clear if the authors propose the descriptor or it was previously established. A couple of figures in the paper (Figs. 10 and 11) need some quality touches, since it is not possible to clearly see the legend to distinguish the lines.

I have some suggestions. Although they are a bit numerous, I classify them as ``minor'', because I think the study is very good and relevant and, as such, it deserves publication. Therefore, this is what they are, suggestions. The authors should feel free to address them or not.

**Mostly minor/editorial-type comments:**

L. 97: Personally, I think you should use numbers when refering to time averages along all the text. So replace two-min by 2-min

L. 158: The exact name of integrated system from Campbell Scientific you are using is the CPEC310. So for me it would be better read something like: "The CPEC310 integrated system from Campbell Scientific comprinsing an ..."

L. 159: Remove "ray" word. IRGA = InfraRed Gas Analyzer

L. 175: replace "Dolton's" by "Dalton's"

L. 185: the CO2 molar mass is 44.01 grams per mole.

L. 218: Mention explicitly the average time windows you employed.

L. 225: In Equation (9) explain what the "i" index refers to

L. 242-244: Why did you use 15d window? What does "i was located" mean?

L. 254: As mentioned previously: It seems not to be very comprehensible. More detail and references need to be added to the Appendix A.2.

L. 303-305: I am not sure if "amplitude" is the appropiate word, because you are refering to the "variation or magnitude" in the diurnal cycle. It might be confused with the amplitudes you got using EMD and spectral analysis.

L. 314: Caption in Fig. 5 replace "donate" by "indicate", or "represent"

L. 355-357: I think it is very important to why the reduction in random error approaches to behavior of white noise.

L. 408: It is necessary to explain what is shown in Figure 10. Explain clearly each one of the predictors (independent variables) you employed in the multiple linear regression. Why does mean $\ln(A\_max)$ and $\ln(P\_max)$? You can exmplain this better in the caprion of the figure.

L. 419-422: Figures 10 and 11. Use the tags ((a), (b), (c), ...) from the plot to better describe the caption of the figure.

L. 428-442: This discussion is not easy to follow.

Appendix:

A.1
L. 689: Equation A.1. In the denominator is missing the "j" index

A.2
L. A figure would be ilustrative to understand what are $P_{d}$, $S_{v}$, etc.

---

## Author Response (AR2)

We have greatly appreciated the reviewer for his/her re-evaluation. His/her reevaluation comments are apparently helpful to finalization of our manuscript for publication. We carefully discussed the comments among our coauthors. Accordingly, the manuscript was revised again in response to the comments below (RC: reviewer's comments; AR: author's response).

**General comments**

This is my second evaluation of this paper. I believe the authors have done a good work for the revision. There are several interesting novelties in the study. There are, nevertheless, some unclear/wrong points, here and there, that in my view, must be revised before publication. Probably, the authors forgot the revision of the abstract left in place the questionable quantification of the storage impact on the overall NEE. Both NEE and Fs were not treated quantitatively as sums in the text, but in the abstract, there is still the mention of "1.9%~4.3% underestimation of the NEE"; which is not supported by presented data. I recommend to simply remove that statement. If I am correct, the annual variation in $CO_2$ concentration in the air column (the global average is around 2-3 ppm) below the eddy covariance system (36 m) should lead to a positive variation in the overall net sink around 0.1 g C $m^{-2}$ $y^{-1}$.

**Response:** Reviewer's understanding matches our thinking. The statement of "1.9%~4.3% underestimation of the NEE" in abstract was removed.

**Specific comments:**

**RC1**: Line 53: "mixing ratio". I prefer the use of the terms "dry molar fraction", and wet molar fraction if water is included.

**AR**: We feel "mixing ratio" is more conventionally used in books (e.g. (Aubinet et al., 2012) and programs (e.g. EasyFlux, Campbell Scientific Inc. UT, US). We have been aware of more uses of "dry molar fraction" in recent literature. At this time, we prefer to keep the mixing ratio unchanged to "dry molar fraction".

**RC2**: Lines 131-133: I cannot understand this sentence.

**AR**: It is a common practice to use a single profile instrumentation for $F_s$ because of cost effective. A single profile system is equipped with one infrared $CO_2/H_2O$ analyzer to avoid the systematic instrument error among the levels that cannot be cancelled if more analyzers are used. Regarding to spatial averaging, AP200 Atmospheric Profile System uses mixing volume technology. The data from this technology actually is temporal average to represent the spatial average.

*Proposed revision:*

Furthermore, time-averaged $[CO_2]$ profiling is employed to represent the $[CO_2]$ average within control volume due to resource constraints. This leads to the gap that the systematic bias and random error in $F_s$ estimate are irreconcilable. (Lines 131-134)

**RC3**: Line 138: "EC site", better EC experimental setup.

**AR**: Revised as suggested.

**RC4**: L154: To avoid confusion, please mention that the EC system is at 36 m above ground. In the description of the experiment would be useful to have an indication of its length (months), and the percentage of gaps due to calibration, filter changes and so on.

**AR**: Thanks for the comments. We have created a table that outlines the instrumentation setups for each flux tower, detailing the configurations for the EC System and the AP200 installation (Table 1). The CPEC310 and AP200 are subject to maintenance and manual calibration on a biannual basis, in the spring and autumn, respectively. The total time required for this process is 16 hours for the CPEC310 and 12 hours for the AP200. Due to calibration, filter changes, and instrument failures, the data from CPEC310 is estimated to be missing approximately 10% per year, while the data from AP200 is estimated to be missing approximately 3% per year.

*Proposed revision:*

Due to calibration, filter changes, and rugged weather, 10% CPEC data and 3% AP200 data were missed in our study period. (Lines 170-172)

**RC5**: L206: Could you add some detail/reference about "empirical modal decomposition"? What are the units of the values reported in Table 2? The same lack of units is repeated at line 362.

**AR**: Thanks for the comments. The empirical mode decomposition (EMD) was employed for the analysis of $[CO_2]$ fluctuations in this study. This analysis comprises of two steps. Initially, the high-frequency $[CO_2]$ time series (10 Hz) is decomposed by EMD, designed to decompose non-linear and non-stationary signals into a set of spectrally independent oscillatory components (Huang and Wu, 2008), the intrinsic mode functions (IMFs). Subsequently, the IMFs proceed through the Fourier spectrum analysis, resulting in a spectrum that preserves local properties in the time domain and provides information in amplitude and frequency domains. This facilitates the identification of hidden local characteristics in the original signal.

We have added a reference about EMD. The unit for $A_m$ is parts per million (ppm), and the unit for $P_m$ is seconds (s). Both Table 2 and the L362 have been revised.

**RC6**: L309: Please check the writing: I would write something like "Significant diurnal variations occurred… as shown in Fig 4."

**AR**: Thanks for the suggestion, we have revised these sentences. The unit for both $\sigma(\varepsilon_s)$ the $|F_s|$ is μmol m$^{-2}$ s$^{-1}$. This study characterizes the relationship between these two variables as a linear model. Consequently, the unit of the intercept of the linear model is μmol m$^{-2}$ s$^{-1}$, while the slope of these two variables is dimensionless.

*Proposed revision:*

Significant diurnal variations and seasonal differences in $F_s$ were observed across the three forest stands, as shown in Fig. 4. (Lines 312-313)

The relationship between $\sigma(\varepsilon_s)$ the $|F_s|$ was characterized by intercepts of 1.99 to 2.82 μmol m$^{-2}$ s$^{-1}$ and slopes of 0.24 to 0.28 (results presented in the Supplementary Tables 5–6). (Lines 365-368)

**RC7**: L554: Deciduous forests exhibit significant variation in LAI, not evergreen forests.

**AR**: Thanks for the comment. The dominant tree species at our study site are deciduous trees. Among them, the Larch plantation forests are mainly composed of two deciduous species: *Larix kaempferi* (Lamb.) Carr. and *Larix olgensis* Henry. Our previous study showed that the leaf area index of the forest stands for all three towers showed significant seasonal variation (Li et al., 2023). We have revised this sentence to make it clearer.

*Proposed revision:*

During the growing season, forests in our study site exhibit higher leaf area index and greater canopy densities than during the dormant season (Li et al., 2023), resulting in longer $P_m$ of short-term [$CO_2$] fluctuations above the canopy (Fig. 3). (Lines 558-561)

**RC8**: L563: Why the energy balance should be determined on by "the valley soil surface"? I believe it is dependent by the difference between incoming and outgoing LW radiation. A source of LW radiation is also at canopy level.

**AR**: We agree with the comment. During night, the loss of heat from the valley soil surface and vegetation canopy through longwave radiation is a primary driver of katabatic flows. To clarify, this sentence has been revised.

*Proposed revision:*

During night, the difference between incoming and outgoing longwave radiation over the valley soil surface and vegetation canopy gives rise to radiative cooling. Subsequently, the air near the soil surface experiences a gravity-induced downslope acceleration, potentially causing katabatic flow. (Lines 567-570)

**RC9**: L567: "diversity", in forest composition/structure or what?

**AR**: Thanks for the comment. We have revised this sentence to make it clearer.

*Proposed revision:*

The terrain unevenness and the complexity of canopy structure significantly affect the airflow divergence in the atmospheric boundary layer. (Lines 584-585)

**RC10**: L610: Where is really the indication of AP200 accuracy?

**AR**: The measurement accuracy of AP200 depends on LI850A infrared $CO_2$-$H_2O$ analyzer (LI-COR Biosciences, NE, USA). Unfortunately, the accuracy in $CO_2$ and $H_2O$ of infrared analyzers from two major manufactures, LI-COR Biosciences and Campbell Scientific, for atmospheric applications has not been specified although the precision, cross-sensitivity, and $CO_2$ and $H_2O$ zero and span drifts are specified (Zhou et al., 2021; Zhou et al., 2022). Zhou et al. (2021; 2022) extensively discussed the accuracies in $CO_2$ and $H_2O$ that are measured using infrared analyzers and defined these accuracies for applications of open-path eddy-covariance systems and closed-path eddy covariance systems to ecosystems. To the best of our knowledge, the accuracies in $CO_2$ and $H_2O$ from LI850 have not been defined for its applications to atmospheric profile measurements. This topic goes beyond the scope of this study.

To better indicate the AP200 accuracy, although unavailable, we added the precision and cross-sensitivity of LI850 while describing the auto zero and span functionality of AP200. The LI-850 exhibits a sensitivity to water vapor of less than 0.1 μmol $CO_2$ per mmol $mol^{-1}$ $H_2O$, and a sensitivity to $CO_2$ of less than 0.0001 mmol $mol^{-1}$ $H_2O$ per μmol $CO_2$.

*Proposed revision:*

The LI-850 analyzer integrated within in AP200 exhibits a sensitivity to water vapor of less than 0.1 μmol $CO_2$ per mmol $mol^{-1}$ $H_2O$, and a sensitivity to $CO_2$ of less than 0.0001 mmol $mol^{-1}$ $H_2O$ per μmol $CO_2$. (Lines 616-619)

**Reference**

Aubinet, M., Vesala, T., and Papale, D.: Eddy Covariance: A Practical Guide to Measurement and Data Analysis, Springer Atmospheric Sciences, Springer, Dordrecht, XXII, 438 pp., 10.1007/978-94-007-2351-1, 2012.

Huang, N. E. and Wu, Z.: A review on Hilbert-Huang transform: Method and its applications to geophysical studies, Reviews of Geophysics, 46, 10.1029/2007rg000228, 2008.

Li, S., Yan, Q., Liu, Z., Wang, X., Yu, F., Teng, D., Sun, Y., Lu, D., Zhang, J., Gao, T., and Zhu, J.: Seasonality of albedo and fraction of absorbed photosynthetically active radiation in the temperate secondary forest ecosystem: A comprehensive observation using Qingyuan Ker towers, Agricultural and Forest Meteorology, 333, 10.1016/j.agrformet.2023.109418, 2023.

Zhou, X., Gao, T., Takle, E. S., Zhen, X., Suyker, A. E., Awada, T., Okalebo, J., and Zhu, J.: Air temperature equation derived from sonic temperature and water vapor mixing ratio for turbulent airflow sampled through closed-path eddy-covariance flux systems, Atmospheric Measurement Techniques, 15, 95-115, 10.5194/amt-15-95-2022, 2022.

Zhou, X., Gao, T., Pang, Y., Mahan, H., Li, X., Zheng, N., Suyker, A. E., Awada, T., and Zhu, J.: Based on Atmospheric Physics and Ecological Principle to Assess the Accuracies of Field $CO_2/H_2O$ Measurements From Infrared Gas Analyzers in Closed-Path Eddy-Covariance Systems, Earth and Space Science, 8, 10.1029/2021ea001763, 2021.